# HybridGS: High-Efficiency Gaussian Splatting Data Compression using Dual-Channel Sparse Representation and Point Cloud Encoder

Qi Yang [1]  Le Yang [2]  Geert Van Der Auwera [3]  Zhu Li [1]

## Abstract

Most existing 3D Gaussian Splatting (3DGS) compression schemes focus on producing compact 3DGS representation via implicit data embedding. They have long encoding and decoding times and highly customized data format, making it difficult for widespread deployment. This paper presents a new 3DGS compression framework called HybridGS, which takes advantage of both compact generation and standardized point cloud data encoding. HybridGS first generates compact and explicit 3DGS data. A dual-channel sparse representation is introduced to supervise the primitive position and feature bit depth. It then utilizes a canonical point cloud encoder to carry out further data compression and form standard output bitstreams. A simple and effective rate control scheme is proposed to pivot the interpretable data compression scheme. HybridGS does not include any modules aimed at improving 3DGS quality during generation. But experiment results show that it still provides comparable reconstruction performance against state-of-the-art methods, with evidently faster encoding and decoding speed. The code is publicly available at https://github.com/Qi-Yangsjtu/HybridGS.

## 1. Introduction

3D Gaussian Splatting (3DGS) (Kerbl et al., 2023) exhibits exceptional capabilities in 3D scene reconstruction, overcoming limitations that previously hindered the practical deployment of real-time radiance field rendering methods.

[1]School of Science and Engineering, University of Missouri-Kansas City, Kansas, US [2]Electrical and Computer Engineering, University of Canterbury, Christchurch, New Zealand [3]Qualcomm, San Diego, US. Correspondence to: Qi Yang <qiyang@umkc.edu>.

*Proceedings of the 42^nd International Conference on Machine Learning*, Vancouver, Canada. PMLR 267, 2025. Copyright 2025 by the author(s).

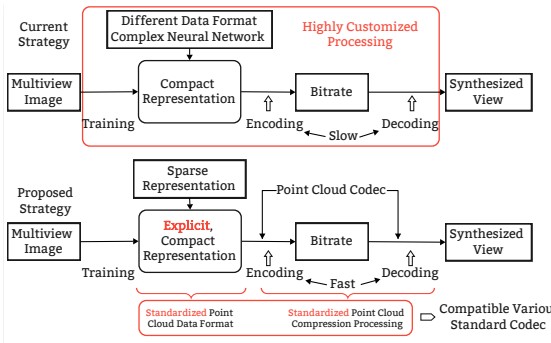

*Figure 1.* Existing generative compression frameworks and the proposed HybridGS.

However, due to the use of the explicit data format and primitive densification strategy, 3DGS has a huge data volume, which is challenging for storage and transmission. 3DGS compression has attracted considerable attention from both industry and academia, which is also the focus of this paper.

In the recent 148-th Moving Picture Expert Group (MPEG) meeting (Liao et al., 2024c), experts from Video Coding (WG4) and Coding of 3D Graphics and Haptics (WG7) reached two consensuses. 1) The generation process of 3DGS has significant room for optimization such as using less primitive, since 3DGS is surjective: two or more distinctly different 3DGS samples may correspond to perceptually close content and quality. This direction will be pursued by WG4 to achieve the compact 3DGS representation, where the input and output can be ground-truth and synthetic views with 3DGS or its variants as intermediate results. 2) Vanilla 3DGS shares the same data format with 3D point clouds. It is reasonable to extend existing point cloud encoders such as the Geometry-based Point Cloud Compression (GPCC) to support the compression of 3DGS data. WG7 suggested that in this case, both the input and output should be 3DGS data. We shall refer to the above two approaches as generative compression methods and traditional compression methods.

In literature, most research efforts have been put into generative compression methods and already achieved impressive compression ratios (Chen et al., 2024b; Liu et al., 2024; Niedermayr et al., 2024; Fan et al., 2024). Meanwhile, traditional compression methods have also been considered in some work (Yang et al., 2024; Huang et al., 2025; Chen

et al., 2024a). There are significant pros and cons associated with both types of techniques. Specifically, with generative compression, superior compression efficiency without noticeable distortion is realizable. However, current generative compression methods normally embed 3DGS primitive information into neural networks or other highly customized data formats, resulting in long encoding and decoding times. In some scenes, the coding latency could become longer than 1 min (see Table 2). Empirical video-on-demand and video live streaming studies indicate that 1s is probably the limit for the user's flow of thought to stay uninterrupted, and 10s is the limit for keeping the user's attention focused on the dialogue (Nielsen, 1994). Besides, to obtain good reconstruction quality over different datasets, some methods need handcrafted selection of hyperparameters or incorporate extra optimization for 3DGS generation, occasionally resulting in compressed data with quality even far exceeding that of the original 3DGS. From the quality evaluation perspective (Yang et al., 2022), the impact of compression then becomes difficult to quantify. These factors make generative compression methods challenging to be standardized and widely deployed (see the upper part of Figure 1).

In contrast, traditional compression methods work with well-defined explicit data formats. With appropriate algorithmic optimizations, real-time encoding and decoding can be achieved, as exemplified by video codecs such as H.264 and HEVC (Sullivan et al., 2012). This suggests that explicit data representations are more conducive to high-speed processing, standardization, and practical deployment. However, due to the huge volume of 3DGS data, the achievable bitrates under lossless compression may still be too large. Lossy compression thus is more attractive but we need to reduce the possible noticeable distortion due to indispensable operations such as quantization (Zaghetto et al., 2024).

With the above observations in mind, we present in this paper a novel hybrid 3DGS compression framework, HybridGS, that takes advantage of both the generative and traditional compression approaches, as shown in the lower part of Figure 1. It leverages the generative compression to produce **explicit and compact 3DGS data**, which have the **same** format as point clouds and are then further compressed using available point cloud encoders. This design has the distinct feature of being able to accelerate both the encoding and decoding processes, typically ranging between 0s and 2s. Thanks to the surjective nature of 3DGS, the operations of lossy processing inherent in the downstream point cloud encoder such as quantization can be taken into account in the generation of 3DGS. This property results in the potential for mitigating the distortion while keeping the feasibility of effectively predicting the quality of the compressed 3DGS.

Correspondingly, HybridGS is implemented using a two-step architecture. In the first step, we pre-define the Bit Depth (BD) of the primitive position and attributes. A dual-channel sparse 3DGS generation scheme is proposed to obtain the desired explicit and compact 3DGS. The dual-channel sparsity here refers to the sparsity of the primitive attributes and position distributions. For compressible 3DGS attributes (i.e., color and rotation), we use the "learnable low-dimensional latent feature + trainable lightweight decoder" to reconstruct them. This can be considered as a generalized Principal Component Analysis (PCA), leading to the desired low-rank representation of compressible attributes with reduced information loss. We integrate a newly proposed differentiable quantization method (Ye et al., 2024) into 3DGS generation to quantize the learnable primitive features. For primitive position, we propose a Learnable Quantizer-based Method (LQM) to generate unique primitives with integer coordinates. Here, owing to the alignment between scene and 3DGS scaling, position de-quantization is not necessarily needed before rendering. The primitive pruning proposed in (Fan et al., 2024) is used to control the primitive number, resulting in two simple but effective rate control strategies. In the second step, GPCC is adopted to produce the standard output bitstream: we split 3DGS data such that the primitive position is compressed in the Octree mode, while other attributes are compressed by RAHT (De Queiroz & Chou, 2016) as normal point cloud attributes.

A significant advantage of the proposed HybridGS is its full compatibility with canonical point cloud encoders, meaning that existing achievements in point cloud compression can be inherited and applied. Compared to state-of-the-art (SOTA) generative compression methods, HybridGS exhibits obviously faster encoding and decoding. To limit the scope of this work and, more importantly, gain interpretability when evaluating compression-related losses, HybridGS is based on the original 3DGS implementation. We deliberately **do not** incorporate any methods to optimize 3DGS generation and improve the reconstruction quality. As such, the upper bound for the reconstruction quality of HybridGS is the vanilla 3DGS. This also implies that optimizations such as Mip-splatting (Yu et al., 2024) targeting 3DGS reconstruction can potentially be included in our method in the future. Our contribution is summarized as follows.

• We develop a new 3DGS compression framework, HybridGS, which takes advantage of both the generative and traditional compression methods.

• We propose a dual-channel sparse 3DGS generation method. The lossy operations in the downstream encoder are considered in the 3DGS generation process, reducing the amount of distortion. Two effective rate control methods are established based on the sparsification strategy.

• Extensive experiments show that HybridGS can provide comparable reconstruction performance against generative compression methods with greatly decreased encoding and

decoding time, as well as flexible rate control capability.

## 2. Preliminaries

In this section, we first summarize the requirement of using point cloud encoders for 3DGS compression coding. Then, two quantization methods for neural networks are presented.

### 2.1. 3DGS Encoding with Point Cloud Encoders

Point cloud encoders can be broadly categorized into two classes, hand-crafted and learning-based. The hand-crafted encoders are studied by MPEG WG7, which have two sub-categories, Video-based Point Cloud Compression (VPCC) and GPCC (Schwarz et al., 2019). VPCC uses existing video codecs to compress the geometry and texture information of a dense point cloud. This is achieved essentially by converting the point cloud into a set of video sequences. GPCC was initially designed to compress highly irregular sparse point cloud samples by exploiting an octree-based encoding strategy. The learning-based point cloud encoders are investigated under MPEG WG7 AI-PCC. UniFHiD (Pang et al., 2024), which is based on SparseConv (Choy et al., 2019), is selected as the study anchor.

Point cloud encoders require that the input data be in the form of integers regardless of they representing geometry or attributes. SparseConv requires point uniqueness: if duplicated geometry points exist, current learning-based methods will directly remove them. In summary, the requirements of using point cloud encoders to compress 3DGS data are: **1)** explicit point-wise primitive description of dimensionality $n \times (3 + M)$, with $n$ primitives, each having $xyz$ coordinates and $M$-channel features; **2)** integer primitive position and feature; and **3)** unique primitive for geometrical positions.

### 2.2. Quantization

**Uniform Quantizer** For a learnable feature $f$ in integers and subject to lossless quantization with respect to $N$-bit BD after training, the simplest is to use Uniform Quantization (UQ) combined with Straight-Through Estimator (STE) (Bengio et al., 2013). Quantization can be formulated as

$$q_i = \left\lfloor \frac{(f_i - f^{min}) \times (2^N - 1)}{f^{max} - f^{min}} + \frac{1}{2} \right\rfloor, \forall i \in 0, ..., S-1, \quad (1)$$

where $S$ is the number of floating values, $\lfloor x \rfloor$ is the floor function, and $f^{max}$, $f^{min}$ are the maximum and minimum possible values. De-quantization is achieved using

$$r_i = \frac{q_i \times (f^{max} - f^{min})}{2^N - 1} + f^{min}, \forall i \in 0, ..., S-1. \quad (2)$$

Considering that the floor function in Equation (1) is not differentiable, STE is used for backpropagating the gradient for training.

**Robust Quantizer** Recently, (Ye et al., 2024) presented a Robust Quantizer (RQ) for neural network quantization that can mitigate perturbation-induced distortion. It has two steps: Perturbation injected Affine Transform for Quantization (PAT-Q) and Denoising Affine Transform for Reconstruction (DAT-R).

●**PAT-Q:** the quantization of a feature $f$ is formulated as

$$q = A(f) + \sigma, \quad A(f) = \frac{f - f_{min}}{f_{max} - f_{min} + \epsilon} \cdot (2^N - 1),$$
$$\sigma = round(A(f)) - A(f), \quad \sigma \in [-0.5, 0.5].$$

$\sigma$ is the perturbation due to UQ, $\epsilon$ is a small constant for preventing division by zero. The introduction of $\sigma$ avoids operations such as clipping, which can degrade performance.

●**DAT-R:** the reconstruction (de-quantization) is cast into a ridge regression problem. The regularization factor $\lambda$ is introduced, and the de-quantization is realized via solving

$$\min_{a,b} \frac{1}{2M} ||a \cdot q + b - f||^2 + \frac{\lambda}{2} a^2, \quad (3)$$

where $M$ is the dimension of $f$. Taking the partial derivatives with respect to $a$ and $b$, and setting the results to zero yield the stationary points, which are

$$a = \frac{Cov_{fq}}{Var_q + \lambda}, \quad b = \bar{f} - a\bar{q}, \quad (4)$$

where $^-$, $Cov$, and $Var$ represent the averaging, covariance, and variance operators. Finally, the reconstruction of the learnable feature is calculated using

$$r = R(q) = a \cdot q + b = \bar{f} + \frac{Cov_{fq}}{Var_q + \lambda}(q - \bar{q}). \quad (5)$$

Equation (5) reflects that the reconstructed feature has a smooth component $\bar{f}$ as well as a non-smooth component $\frac{Cov_{fq}}{Var_q + \lambda}(q - \bar{q})$. By suppressing the non-smooth term, which contributes to training instability, RQ can better balance between the feature fidelity and quantization noise by adjusting $\lambda$. This could facilitate quantization-aware training.

## 3. Method

The framework of HybridGS is depicted in Figure 2. It has two parts: a dual-channel sparse representation module to generate explicit compact 3DGS, and a downstream point cloud encoder to realize encoding and rate control.

### 3.1. Dual-Channel Sparse Representation

#### 3.1.1. Sparse Representation of Attributes

For 3DGS, the sparse representation of attributes can be realized in two aspects: feature channels and precision of feature values. Feature dimensionality reduction is introduced to sparsify feature channels, while quantization is applied to decrease the representation precision of the feature value.

● Feature Channels. Besides the geometry position, vanilla 3DGS has 56 feature channels: 3 channels for color Direct

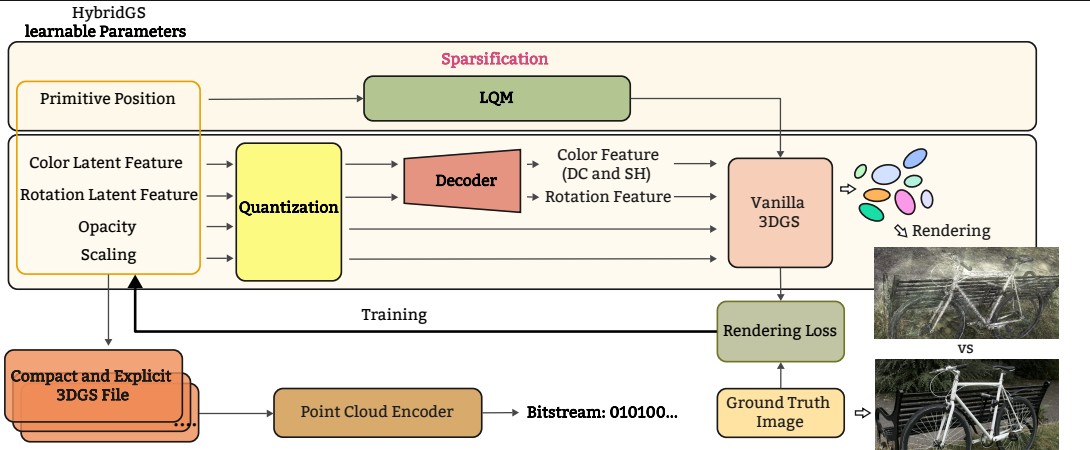

*Figure 2.* Framework of HybridGS.

Current (DC) components, 45 channels for color Spherical Harmonic (SH) coefficients, 1 channel for opacity, 3 channels for scaling, and 4 channels for rotation. Excluding opacity, PCA can be used to partition the remaining 55 features into compressible and compression-vulnerable ones. The concentration of the feature variance in a limited number of principal vectors suggests that a certain feature is compressible. However, using PCA directly in an offline manner leads to loss of high-frequency details and degradation in terms of reconstruction PSNR, as summarized in Appendix A.2. This indicates a training-based approach for representing compressible features should be pursued.

Noting that PCA works as a linear autoencoder (AE) (Lindholm et al., 2022), we employ low-dimensional latent features $f$ and a lightweight decoder $D$ (Girish et al., 2025), a MultiLayer Perceptron (MLP) with one hidden layer, to reconstruct the high-dimensional compressive attributes. In other words, we only constrain the latent feature dimensionality to realize low-rank approximation of features (see Appendix A.2). Mathematically, we have

$$F_r = D_r(f_r) \in R^{n \times 4}, \; f_r \in R^{n \times k_r}, \; k_r < 4, \quad (6)$$

where $F_r$ denotes the 3DGS primitive rotation feature to be compressed. $f_r$ and $D_r$ are the corresponding latent feature and decoder.

PCA analysis indicates that color and rotation attributes are compressible features, while scaling is a compression-vulnerable feature especially for complex samples such as those in the "bicycle" dataset. Therefore, in HybridGS, we use low-dimensional latent features for color and rotation reconstruction (see Equation (6)). Moreover, for color DC and SH, they share the latent feature and the decoder.

●Feature Precision. For vanilla 3DGS, all primitive attributes are in floating-point numbers. Quantization is thus required before the downstream point cloud encoder can be used. Taking quantization into account in 3DGS generation can significantly reduce quality degradation than using it as a post-processing step (see Appendix A.7). We

propose a robust training method inspired by neural network quantization.

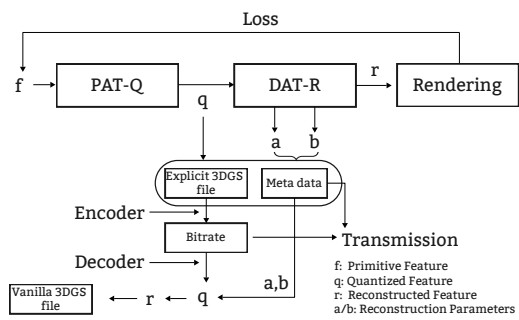

*Figure 3.* Robust Quantization of 3DGS features.

For the latent representations of color and rotation, opacity, and scaling, we initialize a quantizer for each attribute. Using RQ as an example, the training process is shown in Figure 3. The reconstructed features $r$ are used to calculate the rendering loss, while the quantized features $q$ are used for compression and transmission. For RQ, two additional meta data, $a$ and $b$ are required to de-quantize $q$ and compute $r$. A similar technique can be established when employing UQ. We only need to save different meta data for the de-quantization operation. After quantization, we can calculate the number of bits per primitive. Assume that the channel numbers for latent color and rotation features are $k_c$ and $k_r$, and the BD for position, color, opacity, scaling, and rotation attributes are $BD_p$, $BD_c$, $BD_o$, $BD_s$, and $BD_r$. The number of bits per primitive is thus

$$P_{bit} = 3 \cdot (BD_p + BD_s) + k_c \cdot BD_c + BD_o + k_r \cdot BD_r. \quad (7)$$

### 3.1.2. PRIMITIVE SPARSIFICATION

Primitive positions determine the skeleton of 3DGS. More primitives should lead to more detailed texture information but larger data volume. However, (Fan et al., 2024) showed that approximately 60% of the primitives contribute marginally to the reconstruction quality. An improperly large number of primitives can even undermine the repre-

sentational capacity of some primitives (see Appendix A.5). Hence, the sparsification of primitives needs to control primitive number via pruning, and it also has to satisfy the requirement of position uniqueness as specified in Section 2.1.

Primitive position can adopt the same quantization method (UQ or RQ) as other attributes. Here, we additionally design a LQM to generate integer position that can be rendered directly without de-quantization. This corresponds to that the quantized position can illustrate the same content through adjusting the camera position accordingly.

Define the coordinate range as $-2^{N-1} + 1 <= x, y, z <= 2^{N-1} - 1$. Therefore, for different $N$, LQM will generate 3DGS samples with different scales. LQM consists of four steps as shown in the upper part of Figure 4: 3DGS warm-up, 3DGS translation and scaling, primitive position decomposition, and primitive uniqueness and pruning. A naïve 3DGS is first generated after warm-up training based on the vanilla 3DGS. Then, we shift the 3DGS sample to ensure that its Bounding Box (Bbox) center is at the origin. Next, we rescale the 3DGS sample according to the space size circled by BD, in which the primitive position, 3D covariance matrix, and camera position all need to be adjusted (see Appendix A.3). After that, we round the positions coordinates to integers and decompose these integer positions as the inner products of two vectors, namely the basis vector and coding vector (see Appendix A.4). Considering that the basis vector is fixed as $[2^{N-2}, 2^{N-3}, ..., 4, 2, 1]$ given a certain BD, the training reduces to optimizing the coding vector consists of -1, 0, and 1. In each training epoch, we use the basis vector and coding vector to recover the integer position and calculate the loss after rendering.

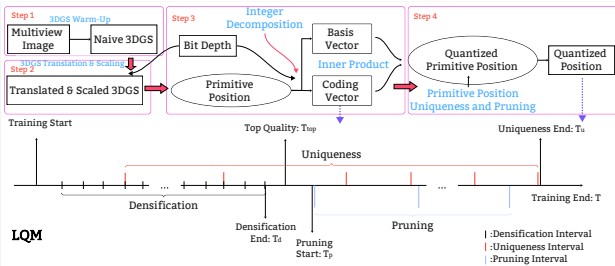

*Figure 4.* Scheme of LQM.

For uniqueness and pruning, a naïve approach is to adopt one time node after 3DGS converges to a stable good performance, where uniqueness and pruning are performed, and then fix the primitive position until the optimization of other attributes pivoting new geometry skeleton is finished. However, in some cases, pruning and uniqueness may remove a few essential primitives, resulting in substantial degradation of the 3DGS quality.

To avoid this, we propose a progressive integer primitive uniqueness and pruning training strategy, which can reduce quality fluctuations and satisfy the rate control for random access. Given the entire training time $T$, four time

nodes are set during 3DGS training: the densification end point $T_d$, the uniqueness end point $T_u$, the pruning start point $T_p$, and the top quality point $T_{top}$. We expect the best reconstruction quality to be reached after densification. We then gradually decrease the number of primitives by removing a certain percentage of less important primitives. The uniqueness ensures that the final 3DGS do not have geometrically duplicated primitive. Therefore, we set $T_d < T_{top} < T_p < T_u \approx T$. The uniqueness operation should be less frequent than the densification in order to prevent immediately reverting the densification result, and we keep the primitive that has the largest size based on scaling attributes. An example of primitive sparsification training is shown in the lower part of Figure 4. Intuitively, $T_{top}$ is the theoretically optimal in terms of reconstruction performance for stopping the training. But after operations like pruning, the quality of 3DGS may not degrade much and in some cases, could even exhibit an improvement.

### 3.2. High-Efficiency Coding

The generated 3DGS samples consist of several parts: an explicit and compact point cloud file, meta data for de-quantization, and two small MLPs for latent features. Meta data and MLPs take several kB, which can be directly stored or transmitted, while the compression is performed on the point cloud file. Considering that GPCC is more stable and effective for point cloud geometry and attribute compression, we use it as the output encoder.

#### 3.2.1. IMPLEMENTATION

Vanilla GPCC supports two types of input, "xyzr" or "xyzrgb". "r" stands for reflectance and "rgb" denotes the color. Both reflectance and color compression are based on RAHT or Predlift. At the implementation level, there are thus two feasible approaches. 1) We may improve the interface of GPCC to allow 3DGS data as input. Primitive position then inherits the compression method of point cloud, while using RAHT/Predlift to compress other primitive attributes. Since the compression of different channels can be treated independently, parallel computing methods can be utilized to accelerate encoding and decoding (e.g., OpenMP (Chandra, 2001)). 2) Considering that the first method needs to modify the internal codebase, another strategy is to divide the generated samples into multiple subsamples with "xyza" or "xyza1a2a3", while "a" represents one channel of primitive attributes. The GPCC can be then used directly. After decoding, we may merge the subsamples together to recover the 3DGS file.

#### 3.2.2. RATE CONTROL

Rate control is important for practical applications. However, with the current generative compression methods,

which use the rate-distortion (RD) optimization as the loss function, the final data rate is difficult to predict. On the contrary, the proposed HybridGS is easier to realize accurate rate control. The bit number of the explicit 3DGS generated in the first step is easy to calculate (see (7)), since we know the primitive number and the assigned bits for each primitive. For the downstream compression, using lossless compression as an example, the compression ratio of the current point cloud encoder is relatively stable. For GPCC, the lossless compression ratio is around 3-4$\times$ for dense point clouds, 2$\times$ for large-scale sparse point clouds. Therefore, we can perform rate control by controlling the size of the explicit 3DGS with downstream compression ratios.

Intuitively, there are three factors that can influence the size of the explicit 3DGS: feature channels, feature BD, and primitive number. For a given scene, we do not know the vanilla data volume before $T_{top}$, which indicates that we need to change these parameters to satisfy the bandwidth requirement after $T_{top}$. Changing feature channels during training results in unstable results and extremely expensive training costs, while changing feature BD and primitive number are more promising for rate control. Therefore, we propose two rate control methods here. We only consider lossless mode of the point cloud encoder for simplicity.

• **Method 1 - Controlling Primitive Number**: the progressive pruning proposed in Section 3.1.2 can realize the soft primitive number control. Given the bandwidth limitation as $B$, the rate control can be formulated as the following optimization problem:

$$\max_{n} \quad Q(GS),$$
$$s.t., R(GS) \leq B \quad \text{and} \quad R(GS) = \frac{n \cdot P_{bit}}{L}, \tag{8}$$

where $GS$ represents the explicit 3DGS samples, $n$ represents the target primitive number, $Q(\cdot)$ and $R(\cdot)$ represent the quality and bitstream of $GS$, and $L$ is the lossless compression ratio of the downstream encoder. The number of primitives in $T_{top}$ is $n_{top}$, the number of primitive need to be pruned is $n_p = n_{top} - n$. Based on the training epoch and the pruning interval $I_p$, the pruning is performed $F_p = \frac{T - T_p}{I_p}$ times. Each time $N' = \frac{N_p}{F_p}$ primitives are pruned until the training ends.

• **Method 2 - Adapting Feature BD**: reducing the feature BD can also lower the bitrate but this is more complex theoretically. In Section 3.1.1, we have demonstrated that features show heterogeneous compressibility. The bandwidth savings are ideally derived from the feature that contributes the least to distortion. Accurate modeling of the distortion sensitivity across different features in 3DGS is currently absent. Hence, we focus on the most straightforward method: bandwidth savings are distributed evenly among all features except for the primitive position as (9), where $\Delta p_{bit}$ is the bits needed to be reduced for each primitive, and $\Delta$ is the BD reduced for each features. A progressive BD reduc-

tion is proposed to mitigate pronounced quality fluctuations: the difference between target $BD_{tar}$ and initial $BD_{ini}$ is $\Delta = BD_{ini} - BD_{tar}$. With disabling pruning, we gradually reduce $BD_{ini}$ with step size 1 until $\Delta = 0$.

$$\max_{\Delta} \quad Q(GS),$$
$$s.t., R(GS) \leq B \quad \text{and} \quad R(GS) = \frac{N \cdot (p_{bit} - \Delta p_{bit})}{L}, \tag{9}$$
$$\Delta p_{bit} = \Delta \cdot (k_c + 1 + 3 + k_r),$$

## 4. Experiment

### 4.1. Experiment Settings

**Datasets**: To comprehensively evaluate the performance of the proposed methods and clearly illustrate the loss caused by compression, we select five scenes from different datasets: "playroom" from deep blending (Hedman et al., 2018), "train" from tanks&temples (Knapitsch et al., 2017), 'bicycle' and "room" from Mip-NeRF360 outdoor and indoor scenes (Barron et al., 2022), and first frame of "Dance_dunhuang_pair" (dance) from PKU-DyMVHumans (Zheng et al., 2024). The partition of training and test images follows the same rule as for the vanilla 3DGS project. The results of overall and other scenes in the above datasets are given in Appendix A.8.

**Comparison methods**: We employ 3DGS as an anchor method and compare both types of representative methods. For generative compression methods, we select Scaffold-GS (Lu et al., 2024), Compact3D (Lee et al., 2024), C3DGS (Niedermayr et al., 2024), CompGS(ECCV) (Navaneet et al., 2025), LightGaussian (Fan et al., 2024), Eagles (Girish et al., 2025), HAC (Chen et al., 2024b), and CompGS(MM) (Liu et al., 2024); for traditional compression methods, we select GGSC (Yang et al., 2024) and HGSC (Huang et al., 2025). The results of these methods are from their original paper or official code.

**Model Parameters**: The BD of the primitive position and other attributes are set to be 16. Training epochs are fixed at 70, 000. $T_d$, $T_p$, and $T_u$ are set as 15, 000, 36, 000, and 66, 000 epochs. We report the results of HybridGS with $k_c = 3/6$ and $k_r = 2$ as for latent representation. RQ ($\lambda = 1e^{-2}$) is used to introduce quantization during training. Other hyperparameters are shown in Appendix A.1. Over the course of pruning, the explicit compact 3DGS will has fewer primitives, resulting in a lower bitrate. We select the samples in the 50, 000 and 70, 000 epochs as High and Low (i.e., pruning 47% and 75% primitives) Rate (HR, LR) points of HybridGS. We use the second implementation in Section 3.2.1 with the "xyza" mode of the GPCC test model v23 (WG7, 2023). The lossless octree and RAHT mode are applied (see Appendix A.7 for results using other encoders).

*Table 1.* Performance of HybridGS. "p" represents the pruning rate of LightGaussian, "$\lambda$" represents the RD loss weighting factor.

| Dataset | | playroom | | train | | bicycle | | room | | dance | |
|---|---|---|---|---|---|---|---|---|---|---|---|
| Type | Metric | PSNR | SIZE (MB) | PSNR | SIZE (MB) | PSNR | SIZE (MB) | PSNR | SIZE (MB) | PSNR | SIZE (MB) |
| Anchor | **3DGS** | **30.03** | **550.67** | **21.89** | **256.73** | **24.49** | **1443.84** | **31.55** | **370.14** | **39.83** | **41.34** |
| Generative Compression | Scaffold-GS | 30.62 | 63.00 | 22.15 | 66.00 | 24.50 | 248.00 | 31.93 | 133.00 | 39.97 | 7.28 |
| | Compact3D | 30.65 | 36.86 | 21.69 | 35.52 | 24.83 | 59.19 | 30.61 | 32.86 | 39.59 | 4.23 |
| | Eagles | 30.32 | 44.00 | 21.74 | 25.00 | 24.91 | 112.00 | 31.84 | 36.00 | 39.89 | 2.93 |
| | C3DGS | 29.89 | 21.66 | 21.86 | 13.25 | 24.97 | 47.15 | 31.14 | 15.03 | 39.67 | 2.42 |
| | CompGS(ECCV) | 30.35 | 10.00 | 21.78 | 12.00 | 25.07 | 29.00 | 31.13 | 9.00 | 39.82 | 1.89 |
| | LightGaussian p=0.66 | 28.36 | 35.64 | 21.50 | 17.17 | 24.51 | 94.65 | 30.75 | 24.32 | 39.05 | 2.90 |
| | LightGaussian p=0.9 | 28.15 | 10.61 | 20.94 | 5.21 | 23.95 | 28.10 | 30.25 | 7.27 | 36.86 | 0.91 |
| | HAC $\lambda=0.0005$ | 30.84 | 6.86 | 22.73 | 12.26 | 25.00 | 44.07 | 31.89 | 8.23 | 39.20 | 0.45 |
| | HAC $\lambda=0.004$ | 30.63 | 3.95 | 22.53 | 7.98 | 24.81 | 26.99 | 31.44 | 5.54 | 38.69 | 0.27 |
| | CompGS (MM) $\lambda=0.001$ | 30.11 | 6.31 | 22.08 | 8.06 | 24.74 | 22.09 | 30.90 | 8.99 | 37.43 | 2.84 |
| | CompGS (MM) $\lambda=0.005$ | 28.98 | 4.89 | 21.78 | 6.21 | 24.43 | 14.62 | 30.35 | 7.16 | 35.35 | 2.69 |
| Traditional Compression | GGSC HR | 29.16 | 260.15 | 18.85 | 50.16 | 19.23 | 224.80 | 29.07 | 166.90 | 33.83 | 14.86 |
| | GGSC LR | 27.30 | 132.24 | 16.64 | 19.30 | 18.31 | 106.36 | 26.42 | 96.03 | 33.42 | 8.30 |
| | HGSC HR | 29.29 | 131.21 | 20.33 | 57.54 | 20.99 | 292.29 | 30.38 | 76.27 | 35.46 | 9.26 |
| | HGSC LR | 28.83 | 96.82 | 19.99 | 41.79 | 20.54 | 223.26 | 29.48 | 57.61 | 34.59 | 7.21 |
| Proposed | HybridGS $k_c=3, k_r=2$ HR | 29.89 | 12.15 | 21.26 | 4.19 | 24.08 | 22.88 | 29.52 | 6.43 | 39.25 | 1.04 |
| | HybridGS $k_c=3, k_r=2$ LR | 29.49 | 5.88 | 20.96 | 2.03 | 23.53 | 11.01 | 29.23 | 3.14 | 37.65 | 0.51 |
| | HybridGS $k_c=6, k_r=2$ HR | 29.89 | 16.08 | 21.49 | 5.63 | 24.10 | 30.21 | 29.75 | 8.28 | 39.31 | 1.32 |
| | HybridGS $k_c=6, k_r=2$ LR | 29.68 | 7.79 | 21.04 | 2.72 | 23.76 | 14.52 | 29.61 | 4.00 | 37.78 | 0.64 |

## 4.2. Experiment Results

### 4.2.1. QUANTITATIVE RESULTS

Table 1 reports the results of HybridGS and other SOTA 3DGS compression methods. We see that: 1) SOTA generative compression methods, e.g., HAC and CompGS(MM), show impressive compression ratios, sometimes with even better PSNR than vanilla 3DGS; 2) SOTA traditional compression methods report significantly larger bitstreams than generative compression methods at similar PSNR, e.g., for "playroom", CompGS(MM) reports 28.98 PSNR with 4.89 MB, while HGSC shows 28.83 PSNR with 96.82 MB. Therefore, only using vanilla 3DGS output as the compression target without modifying the generation process cannot achieve a satisfactory bitstream size; 3) HybridGS can realize a comparable compression ratio with the SOTA generative compression methods with close PSNR, sometimes even better. For example, on "playroom", HybridGS HR has the same PSNR as C3DGS while having a smaller size. On "dance", HybridGS is better than CompGS(MM) in all cases. HybridGS is better than LightGaussian except "room"; 4) for HybridGS, increasing the dimensionality of the latent features can improve PSNR, at the cost of a larger bitstream; 4) RD curves of HybridGS, HAC, and CompGS(MM) on "dance" and "bicycle" are shown in Figure 5. HAC reports a non-strictly monotonic curve on "bicycle" caused by stochasticity, which lies in the rendering nature of 3DGS itself. HybridGS shows slightly higher PSNR than the upper bound on "dance", which is caused by the elimination of redundant primitives via pruning. HybridGS does not outperform HAC in terms of the RD curve, but it offers a notable improvement in encoding and decoding speed, as reported in the next section.

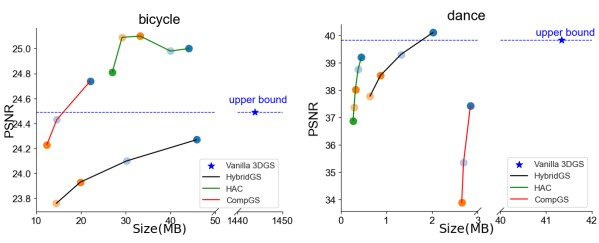

*Figure 5.* RD curve.

### 4.2.2. CODING TIME

One of the advantages of HybridGS is faster encoding and decoding. We use three SOTA methods, i.e., HAC ($\lambda = 0.0005$), CompGS(MM) ($\lambda = 0.001$) and HGSC, as benchmark techniques. LR and HR coding times of HybridGS with $k_c = 3$, $k_r = 2$ are tested, and the results of "bicycle" and "dance" are shown in Table 2. We employ serial encoding and decoding while recording the CPU computation time for HybridGS with GPCC. Data I/O time is excluded. The time for processing primitive position and other attributes is presented separately for the HybridGS LR case. We can see that the coding time of HybridGS is obviously faster than the current SOTA methods. If parallelization techniques are employed to accelerate the GPCC encoder, real-time coding is possible for 3DGS, which is extremely valuable for dynamic 3DGS streaming. We also test de-quantization time $t_{deq}$ and latent decoding time $t_{mlp}$ as preprocessing before rendering. For "bicycle", color $t_{deq}$ and $t_{mlp}$ are around 1s and 0.9s, while 0.6s and 0.001s for rotation. For other attributes, the processing time can be estimated proportionally.

### 4.2.3. RATE ALLOCATION

Table 3 provides a detailed explanation of bitrate distribution. Data in the parentheses denote the size before GPCC. The primary position accounts for 10% of the bitstream, while other attributes exhibit a bitstream roughly proportional to their respective channel numbers. However, the compression ratio of GPCC on position is larger than other attributes, indicating that: 1) RAHT needs to be further improved for 3DGS attributes, and 2) HybridGS generates extremely compact attributes. The size of the latent feature decoders is only related to the feature channels. The quantized metadata, as well as the feature decoders, are very small and can be ignored during rate control.

### 4.2.4. RATE CONTROL

Two different methods for performing rate control are proposed in Section 3.2.2. Four rate points are chosen for "train" and "dance" and the results are presented in Table 4. The

*Table 2.* Coding time.

| Method | bicycle | room |
|---|---|---|
| | Enc/Dec (second) | |
| HAC | 85.03/80.09 | 17.04/15.90 |
| CompGS(MM) | 36.29/22.47 | 7.82/6.28 |
| HGSC HR | 132.32/72.13 | 35.65/16.45 |
| HGSC LR | 124.22/52.94 | 31.64/13.39 |
| HybridGS HR | 1.67/1.77 | 0.44/0.47 |
| HybridGS LR | 0.66/0.92 | 0.26/0.46 |
| -position | 0.09/0.08 | 0.03/0.04 |
| -attribute | 0.57/0.84 | 0.23/0.32 |

*Table 3.* Rate allocation of HybridGS.

| Components | bicycle | dance |
|---|---|---|
| Total Size | 22.88 MB | 1.04 MB |
| -primitive number | 1,286,284 | 56,490 |
| -position | 2.72MB (7.36) | 0.16 MB (0.32) |
| -color (latent feature) | 6.39 MB (7.36) | 0.28 MB (0.32) |
| -opacity | 2.40 MB (2.45) | 0.11 MB (0.11) |
| -scaling | 7.07 MB (7.36) | 0.31 MB (0.32) |
| -rotation (latent feature) | 4.30 MB (4.91) | 0.18 MB (0.22) |
| -metadata | 4 KB | 4 KB |
| -color decoder weights | 13 KB | 13 KB |
| -rotation decoder weights | 4 KB | 4 KB |

*Table 4.* Rate control of HybridGS.

| Dataset | train | | | |
|---|---|---|---|---|
| Type | Method 1 | | Method 2 | |
| Target Rate | PSNR | Real Rate | PSNR | Real Rate |
| 4 MB | 21.13 | 3.96 | 18.20 | 2.20 |
| 6 MB | 21.60 | 5.82 | 21.54 | 5.15 |
| 8 MB | 21.44 | 7.56 | 21.75 | 7.33 |
| 10 MB | 21.54 | 8.59 | 21.75 | 9.86 |
| | dance | | | |
| 0.5 MB | 37.40 | 0.54 | * | * |
| 1 MB | 38.99 | 1.06 | 38.80 | 0.77 |
| 1.5 MB | 39.76 | 1.59 | 39.86 | 1.40 |
| 2 MB | 39.90 | 2.07 | 39.96 | 1.95 |

lossless compression ratio $L$ of GPCC in 3DGS takes values within the range of 1.3 to 1.5 based on preliminary experiments, we set $L = 1.3$ here. We see that: 1) in most cases, method 1 demonstrates a more precise approximation to the target bitrate compared to method 2; 2) "*" in "dance": method 2 means that we cannot achieve the target rate solely by adapting BD. The primitive position rate is larger than the target rate, resulting in attribute BDs reduced to 0; 3) method 1 reports larger rate error for 10 MB of "train" than other rates. The reason is that different primitive densities will influence the lossless compression ratio. For GPCC, the denser the point cloud, the higher the compression ratio. An improvement of the proposed rate control strategy is to first calculate the point density to obtain a more accurate estimation of $L$; 4) $L$ on 3DGS is lower than the traditional point cloud. The geometric distribution of 3DGS is characterized by local density, global sparsity, and the presence of solid regions, distinguishing it from traditional dense point clouds that are hollow with only surface representations. There is a room to further optimize GPCC on 3DGS compression. We also find after $T_p$, although we allow optimize primitive position, LQM will not generate duplicated primitive position at most cases because 3DGS is very sparse, so uniqueness will not influence rate control methods proposed in Section 3.2.2.

*Table 5.* HybridGS with different BDs and quantizers.

| Dataset | train | | | | | | | |
|---|---|---|---|---|---|---|---|---|
| Type | RQ | | | | UQ | | | |
| Rate | High Rate | | Low Rate | | High Rate | | Low Rate | |
| BD | PSNR | SIZE | PSNR | SIZE | PSNR | SIZE | PSNR | SIZE |
| 12 | 20.06 | 1.54 | 19.71 | 0.74 | 19.95 | 1.45 | 19.58 | 0.70 |
| 13 | 20.95 | 2.50 | 20.48 | 1.21 | 20.72 | 2.50 | 20.37 | 1.22 |
| 14 | 21.14 | 3.37 | 20.95 | 1.62 | 21.20 | 3.48 | 20.80 | 1.71 |
| 15 | 21.30 | 4.96 | 20.96 | 2.40 | 21.42 | 5.09 | 20.92 | 2.46 |
| 16 | 21.49 | 5.63 | 21.04 | 2.72 | 21.66 | 5.60 | 21.24 | 2.89 |
| | dance | | | | | | | |
| 12 | 39.23 | 0.84 | 37.58 | 0.41 | 39.32 | 0.84 | 37.69 | 0.41 |
| 13 | 39.28 | 0.99 | 37.59 | 0.48 | 39.35 | 0.93 | 37.59 | 0.45 |
| 14 | 39.50 | 1.16 | 37.86 | 0.56 | 39.33 | 1.14 | 37.92 | 0.56 |
| 15 | 39.43 | 1.20 | 37.91 | 0.58 | 39.49 | 1.25 | 37.93 | 0.61 |
| 16 | 39.31 | 1.32 | 37.78 | 0.64 | 39.35 | 1.37 | 37.85 | 0.66 |

### 4.2.5. INFLUENCE OF BD AND QUANTIZER

Table 5 reports the performance of HybridGS with different BDs on "train" and "dance". 12 to 16 BDs are tested with $k_c = 3$, $k_r = 2$. We can see that: 1) with the increase of BD, HybridGS generally presents better PSNR and larger size; 2) for "train", we can use BD larger than 16 to further approach the upper bound; 3) based on "dance", when reaching a certain threshold, only increasing BD will not

improve PSNR. It indicates that selecting a proper BD can facilitate saving bandwidth. For different quantizers, we can see that: 1) for "train", RQ reports higher PSNR under low BD cases; 2) for "dance", RQ and UQ have close performance; 3) "train" is more challenging than "dance", therefore we believe that RQ is a more robust method for complex content and low bitrate conditions.

## 5. Related Work

Our work is inspired by previous efforts in 3DGS compact generation and 3DGS data compression. In this section, we will first briefly introduce the achievement of 3DGS compact generation in academia, then summarize the explorations of 3DGS data compression within MPEG.

### 5.1. 3DGS Compact Generation

Multiple methods have been proposed to produce a more compact 3DGS, in which additional constraints are injected into the training process to affect and control data generation. Considering that excessive primitive growth is one of the main reasons for the huge data volume, CompGS(ECCV) (Navaneet et al., 2025) and C3DGS (Niedermayr et al., 2024) proposed using the codebook to restrict the number of primitives during densification, which can be regarded as the pioneer of the 3DGS compact representation. Scaffold-GS (Lu et al., 2024) realized structured GS generation by introducing MLP during training, in which data are implicitly stored in MLP and significantly reduced storage and memory cost. Based on Scaffold-GS, HAC (Chen et al., 2024b) proposed using multiresolution hash coding to train a context model, which can realize approximately $60 \times$ reduced storage. CompGS(MM) (Liu et al., 2024) shared a close concept with Scaffold-GS, in which a group of anchor primitives is selected to realize inter-primitive prediction. They also formulated a rate-constrained optimization to balance the quality and bitrate. LightGaussian (Fan et al., 2024) proposed a Gaussian pruning strategy, followed by a SH distillation and vector quantization. Due to fewer primitives and lower SH dimensions, LightGaussian realizes a $15 \times$ reduction and 200+ FPS. ContextGS (Wang et al., 2024) designed an autogressive model that encodes GS primitives with multiple anchor levels, which achieved a higher compression ratio than HAC. Based on these studies, MPEG WG4 (Liao et al., 2024b) decided to add 3DGS to the work-

ing draft and standardize its compact generation method. Several exploratory experiments are presented and discussed in the 147-th MPEG meeting, such as performance analysis of LightGaussian compression using neural network compression (Kim et al., 2024) and 3DGS for light field video transmission (Kawai & Nakagami, 2024). The industry held an optimistic outlook on the practical applications of 3DGS, advancing the standardization process of 3DGS compression and streaming.

### 5.2. 3D GS Data Compression

The vanilla 3DGS primitive consists of multiple attributes: position, DC color, SH coefficients, opacity, scale, and rotation. It can be regarded as a sparse point cloud with high-dimension point features. Inspired by the point cloud compression method, GGSC (Yang et al., 2024) is the first traditional 3DGS compression benchmark, which is based on graph signal processing and uses two branches to compress the primitive center and other attributes. At the same time, MPEG WG7 established a joint EE with WG4 to explore how to make 3DGS compatible with the current 3D data compression codec, such as GPCC (WG7, 2023), which is the so-called traditional compression method. Some preliminary experiments have been conducted base on current GPCC codec. The proposal (Fujii et al., 2024) treated each 3DGS as a point cloud with attributes of opacity, size, quaternion, and SH coefficients for coding. They used Octree to code the center and RAHT (De Queiroz & Chou, 2016) to code other attributes. All attributes are quantized by being multiplied by 1000 and then converted to an integer. The conclusion is that the current GPCC implementation is not effective enough to deal with GS attributes. To better analyze the reason, proposal (Zaghetto et al., 2024) explored the influence of center quantization. The conclusion is that the optimal quantization BD is around 14 to 18 BD corresponding to different datasets: 16 to 18 BD for Mip-NeRF360 (Barron et al., 2022), 18 BD for Tanks&Temples (Knapitsch et al., 2017), and 14 BD for Deep Blending (Hedman et al., 2018). Compared with general point cloud samples used in the PCC study, 3D GS samples require a higher BD. Therefore, 3DGS center quantization introduces quite a large distortion corresponding to the current PCC method.

Besides GPCC, Learning-based Point Cloud Compression (LPCC) framework studied in MPEG AI-PCC (WG2 & Requirements, 2024) should theoretically also be able to handle 3DGS data. The requirement of adding radiance field coding as a new section for MPEG AI has been proposed (Liao et al., 2024a). SparseConv (Choy et al., 2019) is the key technology used in the current SOTA LPCC framework, such as PCGC (Wang et al., 2021) and SparsePCGC (Wang et al., 2022). However, SparseConv requires that the spatial coordinates be integers, and each spatial location must contain only one primitive. Vanilla 3D GS allows multiple

primitives to occupy the same spatial location; therefore, primitive uniqueness is a necessary step after quantization before using the SOTA LPCC method. Based on the results in Appendix A.7, it will incur severe quality loss. Other related topics were also studied: (Gao et al., 2024) compared the 2D rendering loss and 3D reconstruction supervision for 3D GS compression, they found 2D rendering loss is more effective than point-wise point cloud quality metric (i.e., point-to-point). Although SOTA point cloud metrics (Meynet et al., 2020; Yang et al., 2022; Zhu et al., 2024; Yang et al., 2023) have not been tested, it indicates that 3D GS requires specialized quality assessment studies with respect to lossy compression.

Taking into account the above evidence, using the point cloud codec to compress 3DGS data deserves commensurate attention and technological improvement. Therefore, in this paper, we solve the problem of 3DGS quantization and primitive uniqueness, serving as a catalyst for new research endeavors on 3DGS compression.

## 6. Conclusion

This paper presents a new 3DGS compression method HybridGS, which first generates an explicit and compact 3DGS file and then uses canonical point cloud encoders to realize high-efficient coding and flexible rate control. HybridGS has dual-channel sparse representation during 3DGS generation, including feature dimensionality reduction, quantization, and progressive primitive position control. HybridGS reports comparable performance with SOTA methods and faster coding and decoding, as well as demonstrating characteristics of interpretability, compatibility, and alignment with the demands of standardization.

**Limitations:** The optimal compression efficiency of HybridGS is lower than end-to-end generation compression methods using RD loss (Liu et al., 2024) as supervision.

**Future Work:** HybridGS requires hyperparameters like latent feature dimension and BD. An adaptive parameter selection algorithm may help HybridGS achieve a better quality and size tradeoff. For rate control, different from reducing the BD and primitive pruning, decreasing the latent feature dimension during 3DGS generation can lead to feature space collapse, as well as training a new decoder with extra time. How to achieve smooth latent feature dimension reduction without significantly affecting the rendering quality is thus a research topic worthy of investigation. Besides 2D rendering loss, explicit 3DGS generation might benefit from new loss functions in 3D space such as Chamfer distance to optimize primitive distribution to improve point cloud encoder efficiency (Yang et al., 2023).

## Acknowledgments

This work is supported in part by an award from NSF 2148382, and a gift grant from Qualcomm.

## Impact Statement

This paper presents work whose goal is to advance the field of 3D Gaussian Splatting Compression and Machine Learning. There are many potential societal consequences of our work, none which we feel must be specifically highlighted here.

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

# A. Appendix

## A.1. Model Parameters

For LQM, before generating the naïve 3DGS, we first conduct outlier removal to clean the output of COLMAP, facilitating the following 3DGS translation and scaling (see Appendix A.3). We use the function "remove_statistical_outlier" from Open3D (Zhou et al., 2018) to realize outlier removal. Two parameters used in this function are set as "nb_neighbors = 50" and "std_ratio = 2.0" as suggested by the official documents. The first $t_1 = 7,500$ epochs are used for 3DGS warm up, followed by 3DGS translation and scaling. After 3DGS primitive position decomposition, Adam optimizer is used for coding vector training with initial learning rate of $1 \times 10^{-5}$. The initial learning rate of the scaling attribute is set to $0.2 \times \log(k) \times \text{scaling\_lr}(t_1)$ after 3DGS scaling, where $\text{scaling\_lr}(t_1)$ is the original scaling learning rate in $t_1$ epochs. The uniqueness, densification and pruning intervals are 500, 100, and 2500. For uniqueness, it is activated from 7,500 to 66,000 epochs. After 66,000 epochs, we fix the primitive position and update only other attributes. For the position that has more than one primitives, we keep the primitive that has the largest size based on scaling attributes, other primitives are removed directly (given a primitive with scale $[S_x, S_y, S_z]$, the size are defined as $V = S_x \times S_y \times S_z$). For pruning, each time we remove 0.1% primitives. For color and rotation latent feature decoders, we use a single-layer MLP with 50 hidden units and ReLu as the activation function. The learning rate of latent features and decoders for color are 0.001 and 0.001, while 0.005 and 0.015 for rotation. Other parameters are the same as in vanilla 3DGS. All the experiments are tested on Intel Core i9-14900HX, NVIDIA RTX 4090 Laptop.

## A.2. PCA Analysis and Learnable Low-Rank Approximation

We randomly select 10, 000 primitives from the "bicycle" and "dance" datasets. The PCA results for color, scaling and rotation attributes are shown in Figure 6. We can see that: 1) for color, a significant portion of the feature energy is concentrated within the first 20 principal components under both "bicycle" and "dance"; 2) for scaling, the energy is more evenly distributed under "bicycle", while under "dance", most energy is in the first principal component. These results indicate that the scene characteristics affect the scaling attribute distribution greatly. Considering that the scaling features are not always compressible and they have three channels only, we do not attempt to extract their latent representation; 3) for rotation, most energy is concentrated within the first two principal components. Therefore, we use $f_r = 2$ in Section 4.

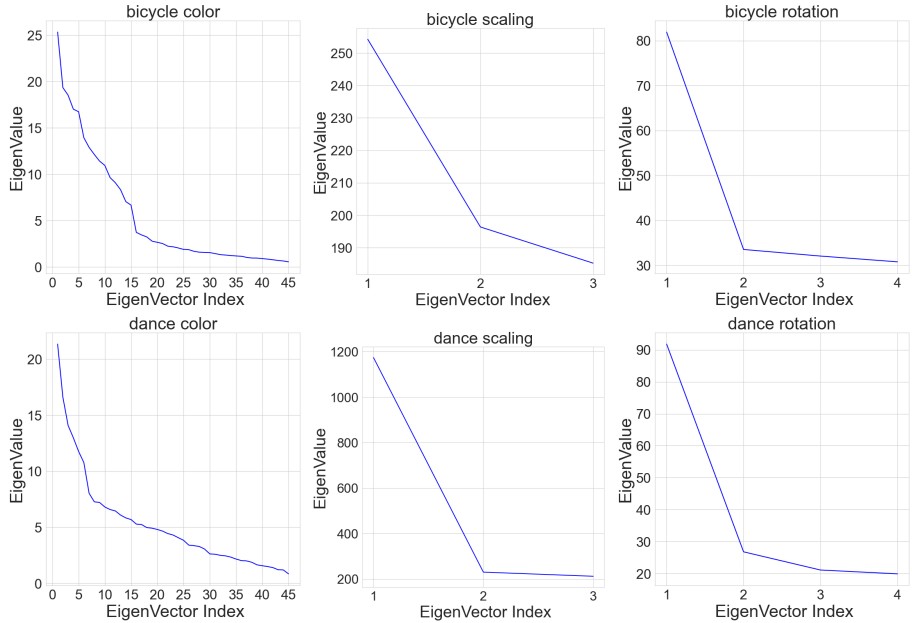

*Figure 6.* PCA results of "bicycle" and "dance".

We use data from "dance" to illustrate the difference between using PCA and the proposed learnable low-dimensional latent features with lightweight trainable decoder. After obtaining vanilla 3DGS samples, we perform dimensionality reduction

via PCA for color and rotation attributes with the first 6 and 2 principal components retained, which is then followed by reconstruction. Meanwhile, considering that PCA works as a linear auto-encoder (AE), we use an one-hidden-layer MLP without nonlinear activation functions such as ReLu and re-train HybridGS with $k_c = 6$ and $k_r = 2$.

Table 6. PSNR comparison under different compressive latent feature extraction methods.

| Method | PSNR |
|---|---|
| vanilla 3DGS | 39.83 |
| PCA | 21.50 |
| HybridGS (Linear Decoder) $k_c = 6, k_r = 2$ | 39.27 |
| HybridGS (with ReLu) $k_c = 6, k_r = 2$ | 39.31 |

We see that the PSNR of the proposed HybridGS with trainable latent feature and decoder is obviously higher than directly applying PCA. The reason is that making both the latent feature and decoder learnable under the supervision of the reconstruction loss enables a better low-rank 3DGS feature representation, as explained below.

Using PCA to realize dimensionality reduction and reconstruction is equivalent to a linear AE in the sense that the objective is to learn a low-dimensional representation $z \in R^{q \times 1}$ of the 3DGS feature $x \in R^{p \times 1}$, where $q < p$. The associated encoder and decoder can be formulated as

$$z = f_\theta(x) = W_e x + b_e, \quad x = h_\theta(z) = W_d z + b_d. \tag{10}$$

By the minimization of the squared reconstruction error, we have, with $\theta = \{W_e, W_d\}$,

$$\hat{\theta} = arg \min_\theta \sum_i^n ||x_i - (W_d W_e x_i + W_d b_e + b_d)||^2, \tag{11}$$

Since $b_e$ is a free parameter and the encoder mapping simplifies to $z = W_e x$, the optimal value for $b_d$ is

$$b_d = \frac{1}{n} \sum_{i=1}^n (x_i - W_d W_e x_i) = (I - W_d W_e)\overline{x}, \tag{12}$$

where $\overline{x} = \frac{1}{n} \sum_{i=1}^n x_i$. The objective Equation (11) thus simplifies to

$$\hat{W}_e, \hat{W}_d = arg \min_{W_e, W_d} \sum_{i=1}^n ||x_{0,i} - W_d W_e x_{0,i}||^2,$$
$$= arg \min_{W_e, W_d} ||X_0 - \hat{X}_0||_F^2, \tag{13}$$

where $x_{0,1} = x_i - \overline{x}$, $X_0 = [x_{0,1}, x_{0,2}, ..., x_{0,n}]$ and $\hat{X}_0 = [\hat{x}_{0,1}, \hat{x}_{0,2}, ..., \hat{x}_{0,n}]$ with $\hat{x}_{0,i} = W_d W_e x_{0,i}$ is the reconstruction of the centred $i$th data point. PCA can be considered as finding the optimal rank-$q$ approximation $\hat{X}_0$ of the centered data matrix $X_0$ in the sense of Frobenius norm minimization. Applying the singular value decomposition (SVD) to $X_0$ results in $X_0 = U\Sigma V^T$. Using a partitioned matrix notation, we have

$$U = [U_1 \ U_2], \quad \Sigma = \begin{bmatrix} \Sigma_1 & 0 \\ 0 & \Sigma_2 \end{bmatrix}, \quad V = [V_1 \ V_2] \tag{14}$$

where $U_1 \in R^{p \times q}$, $\Sigma_1 \in R^{q \times q}$, and $V_1 \in R^{n \times q}$. Therefore, the best rank-$q$ approximation of $X_0$ is then obtained by replacing $\Sigma_2$ by a zero matrix, and consequently,

$$\hat{X}_0 = U_1 \Sigma_1 V_1^T = W_d W_e X_0. \tag{15}$$

Choosing $W_e = U_1^T$ and $W_d = U_1$ attains the desired result:

$$W_d W_e X_0 = U_1 U_1^T [U_1 \ U_2] \begin{bmatrix} \Sigma_1 & 0 \\ 0 & \Sigma_2 \end{bmatrix} \begin{bmatrix} V_1^T \\ V_2^T \end{bmatrix} = U_1 \Sigma_1 V_1^T. \tag{16}$$

The encoder thus represents the input data $X$ in a low-dimensional space using $Z = W_e X = U_1^T X$.

More generally, the low-dimensional representation $Z'$ may be found via solving

$$\min_{\theta'} \quad \|Z' - U_1^T X\|$$
$$s.t. \ Z' = f'_{\theta'}(X). \tag{17}$$

where $f'_{\theta'}(\cdot)$ is an encoder having both linear and nonlinear transformations. Introducing nonlinear transformations into the encoder may be able to improve the reconstruction performance, while the use of 2-norm loss function should tend to push the majority of the energy into a low-dimensional feature space. However, the above dimensionality reduction typically induces loss of information, particularly in the high frequency region.

Considering that 3DGS data $X$ are surjective, we may drop the PCA-based compression projection $U_1^T$ and directly search for, through training, a decoder $f_{\theta''}(Z'')$ that maps an unknown but learnable low-dimensional feature matrix $Z''$ to reconstruct $X$. Mathematically, we are going to solve

$$\min_{\theta'', Z''} \quad \|f_{\theta''}(Z'') - X\|. \tag{18}$$

Ideally, if $X$ truly has rank $q$ (i.e., the SVD of $X$ has only $q$ non-zero singular values), we should be able to losslessly recover $X$ from $Z''$ using even a linear decoder, as can be done with PCA. When the decoder $f_{\theta''}(Z'')$ is further equipped with nonlinear activation functions, improved performance can be obtained, as shown empirically in Table 6, as $Z''$ may contain significantly more information than the linear operation-based PCA.

### A.3. 3DGS Translation and Scaling

The naïve 3DGS sample $\mathcal{G}'$ might be located in any position in the 3D space due to the random position of initial point cloud $\mathcal{P}$. Limiting the primitive positions between $-2^{N-1} + 1$ and $2^{N-1} - 1$ requires first shifting and then rescaling the Bbox of $\mathcal{G}'$, as shown in Figure 7.

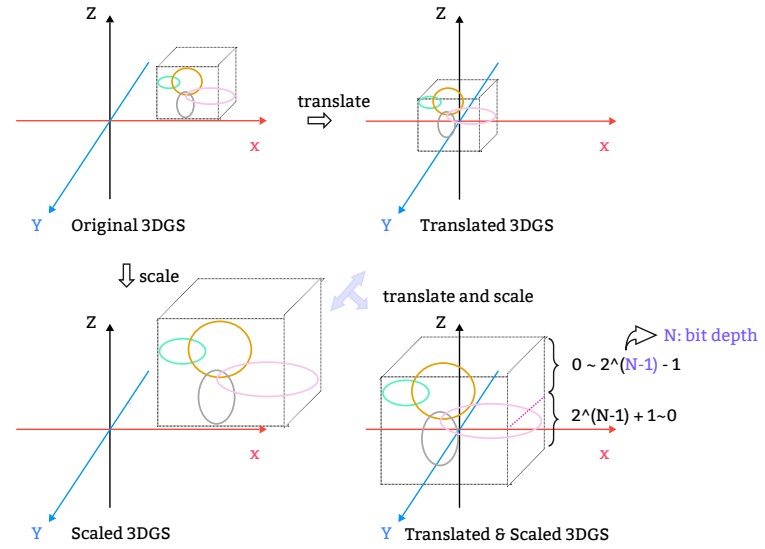

*Figure 7.* 3DGS translation and scaling.

• 3DGS translation: for a 3DGS $\mathcal{G}'$ with $m$ primitives and the $i$th primitive position being $\text{pc}_i = [x_i, y_i, z_i]$. The Bbox center of $\mathcal{G}'$ is $\mathcal{C} = \frac{1}{2}[(x_i)^i_{max} + (x_i)^i_{min}, \ (y_i)^i_{max} + (y_i)^i_{min}, \ (z_i)^i_{max} + (z_i)^i_{min}] = [\delta X, \delta Y, \delta Z]$. Shifting $\mathcal{G}'$ to the origin can be realized by $\text{pc}_i = \text{pc}_i - \mathcal{C}, i \in [1, m]$.

• 3DGS scaling: for a 3D Gaussian primitive $\mathcal{G}'(x) = e^{-\frac{1}{2}(x-\mu)^T \Sigma^{-1}(x-\mu)}$, where $\mu$ is the point mean and $\sum$ is the 3D covariance matrix, scaling $\mathcal{G}'(x)$ with $k$ requires scaling the point position and covariance matrix at the same time: $y = kx$, $\sum' = k^2 \sum$. $y = kx$ is easy to understand, we provide a proof of $\sum' = k^2 \sum$ here.

*Proof.* Assuming $\mathcal{G}''(y)$ is the scaled $\mathcal{G}'(x)$ with $y = kx$,

$$\mathcal{G}''(y) = \mathcal{G}''(kx) = \mathcal{G}'(\frac{1}{k}y) = e^{-\frac{1}{2}(\frac{y}{k}-\mu)^T \sum^{-1}(\frac{y}{k}-\mu)}$$

$$= e^{-\frac{1}{2k^2}(y-k\mu)^T \sum^{-1}(y-k\mu)} \tag{19}$$

$$= e^{-\frac{1}{2}(y-\mu')^T \sum'^{-1}(y-\mu')}$$

where $\mu' = k\mu, \because k^{-2} \sum^{-1} = \sum'^{-1}, \therefore \sum' = k^2 \sum.$  $\square$

For the scaling ratio $k$, we set $k = \frac{2^{N-1}-1}{\text{pc}_{max}}$, $\text{pc}_{max} = max(|x^i|_{max}, |y^i|_{max}, |z^i|_{max})$, $|x^i|_{max}$ represents the maximum absolute value of the $x$ coordinates, and the same for $y$ and $z$ coordinates. 3DGS translation and scaling will not change the rendering quality because it does not change the relative position and overlapping of the primitives. The camera positions also need to be adjusted accordingly after the 3DGS translation and scaling.

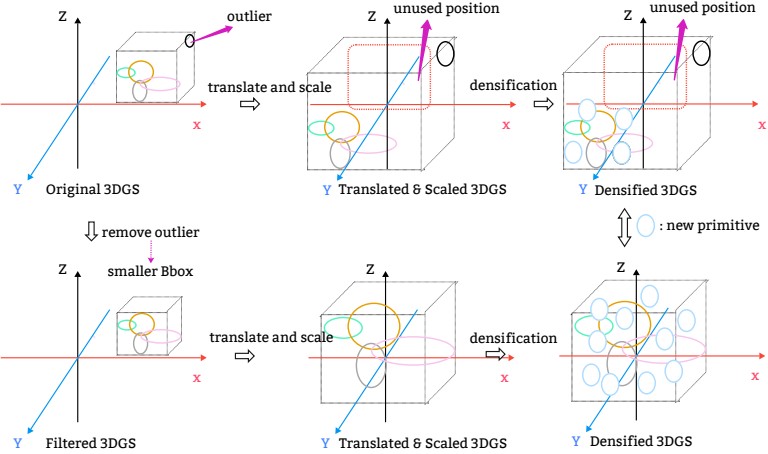

*Figure 8.* Influence of outlier removal.

To take full advantage of finite space, an outlier removal is applied before initializing the point cloud to a 3DGS $\mathcal{G}$, i.e., $\mathcal{G} = \mathcal{I}\{O(\mathcal{P})\}$, $O(\cdot)$ is the outlier removal algorithm and $\mathcal{I}\{\cdot\}$ is the initialization function mapping point cloud to a 3DGS. Outlier removal will influence the results of GS translation and scaling. Toy examples are used for illustration in Figure 8.

Figure 8 shows the influence of outlier removal on 3DGS translation and scaling. The sparse point cloud generated by COLMAP might have a lot of noisy scatter points that are far away from the target reconstruction content. The noisy points, which will incur an oversized Bbox for the initialized 3DGS, can cause spatial waste and primitive densified in a relatively small Bbox after GS scaling. It consequently reduces the resolution of 3DGS in the following quantization learning process, and results in suboptimal visual quality.

Both outlier removal (s1) and 3DGS translation (s2) are designed to improve space utilization in LQM. To highlight the effectiveness of these two modules, we evaluate the performance of LQM on "truck " by disabling either or both of these two modules. The results are shown in Figure 9. To highlight the influence of s1 and s2, we only consider uniqueness in this section and other compression operation, such as dimensionality reduction, feature quantization, and primitive pruning are disabled. We train 30000 epochs as vanilla 3DGS. We see that: 1) these two modules show more obvious performance improvement in low BD, considering that low BD offers less available space; 2) the impact of 3DGS translation is less prominent than that of outlier removal. We think the results are not always stable: the effectiveness of 3DGS translation has a strong correlation with the spatial position of initialized 3DGS, which may be highly random.

### A.4. Primitive Position Decomposition

Given a primitive position $\text{pc}_i = [x_i, y_i, z_i]$ after 3DGS translation and scaling, whose coordinates are floating numbers and $-2^{N-1} + 1 <= x_i, y_i, z_i <= 2^{N-1} - 1$, a rounding function is first applied to calculate the nearest integer, i.e., $\mathcal{R}(\text{pc}_i)$. Then, the integer is represented by a $N$-bit encoding, which is actually the inner product between a basis vector $\mathbf{e} = [2^{N-2}, 2^{N-3}, ..., 4, 2, 1]^T \in \text{R}^{(N-1)\times 1}$ and a coding vector $\mathbf{t} = [t_1, t_2, ..., t_{N-1}]^T \in \text{R}^{(N-1)\times 1}$ with $t_j \in \{-1, 0, 1\}$,

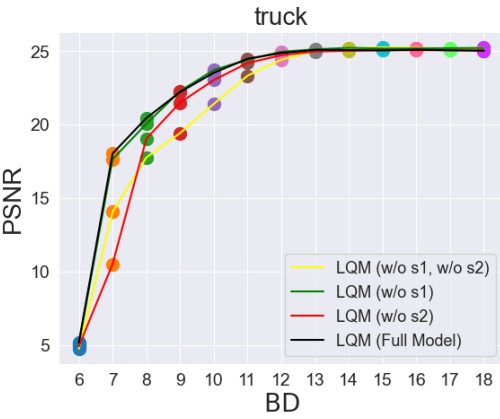

*Figure 9.* Ablation study on outlier removal and 3DGS translation.

i.e., $\mathcal{R}(\mathrm{pc}_i) = [<\mathbf{e}_x^i, \mathbf{t}_x^i>, <\mathbf{e}_y^i, \mathbf{t}_y^i>, <\mathbf{e}_z^i, \mathbf{t}_z^i>]$. Considering that the primitive positions share the same BD and the same basis vector, using $x$ coordinate as an example, the primitive position matrix can be expressed in the following matrix form

$$\mathcal{R}(\mathrm{pc})_x = \mathbf{L}\mathbf{e}, \tag{20}$$

where $\mathbf{L} = [\mathbf{t}_x^1, \mathbf{t}_x^2, ..., \mathbf{t}_x^m]^\mathrm{T}$ and $\mathbf{t}_x^{i\,\mathrm{T}} = [\mathrm{t}_1, \mathrm{t}_2, ..., \mathrm{t}_{N-1}]_x^i$.

In the vanilla 3DGS generation, $\mathrm{pc}$ is the learnable parameters that are updated by gradient descent derived from the rendering loss. After primitive position decomposition, $\mathbf{L}$ replaces $\mathrm{pc}$ and is updated during training. To ensure $\mathbf{L}$ is a coding vector of finite integers, we limit the value range and a rounding operation is used during training with the STE to realize gradient propagation.

### A.5. Influence of Pruning

To generate 3DGS samples with different rates, we introduce primitive pruning at iteration $T_p = 36,000$. The influence of pruning is shown in Figure 10. We can see that: 1) the decrease in the number of primitives results in a general deterioration of reconstruction quality; 2) for the low rate of "bicycle", the testing PSNR reports a stable value, while training PSNR exhibits increased variability. It indicates that there is an inconsistency in the trends of reconstruction quality between training views and testing views. Considering that the final quality of experience is influenced by both training and testing views, we recommend future 3DGS compression studies pay more attention to training views rather than only reporting quality results on testing views.

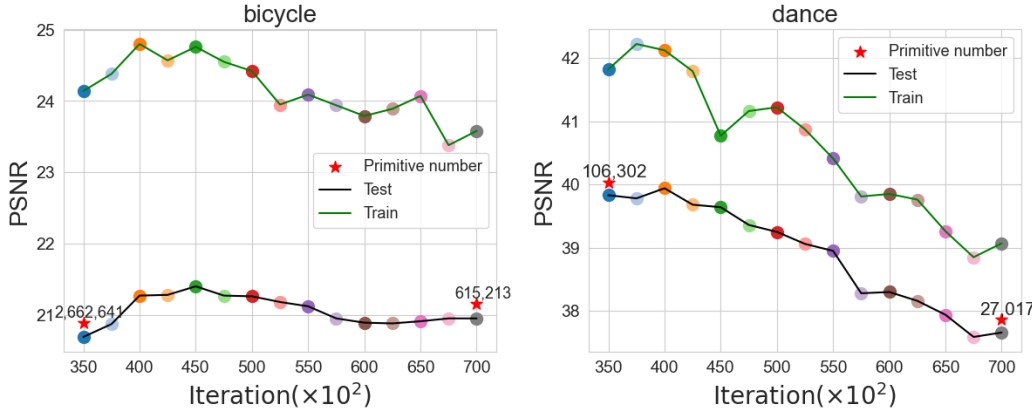

*Figure 10.* Iteration vs. PSNR curves of HybridGS with pruning.

*Table 7.* Ablation study on primitive position uniqueness.

| Dataset | | drjohnson | | | room | | |
|---|---|---|---|---|---|---|---|
| BD | Method | PSNR | Size (MB) | PN(K) | PSNR | Size (MB) | PN(K) |
| 10 | LQM | 28.51 | 156.94 | 663 | 28.98 | 47.03 | 198 |
| | w/o | 28.72 | 552.14 | 2334 | 30.16 | 284.35 | 1202 |
| | UQ | 28.80 | 151.39 | 640 | 30.21 | 78.70 | 332 |
| | w/o | 28.61 | 578.35 | 2445 | 30.23 | 270.26 | 1143 |
| 12 | LQM | 28.87 | 311.89 | 1318 | 31.00 | 136.45 | 576 |
| | w/o | 28.85 | 548.04 | 2317 | 30.63 | 281.54 | 1190 |
| | UQ | 29.09 | 331.53 | 1401 | 31.06 | 143.76 | 608 |
| | w/o | 29.04 | 579.36 | 2449 | 31.16 | 277.46 | 1173 |
| 14 | LQM | 28.90 | 361.29 | 1527 | 31.20 | 200.18 | 846 |
| | w/o | 28.84 | 550.52 | 2327 | 31.34 | 281.96 | 1192 |
| | UQ | 29.04 | 438.00 | 1851 | 31.38 | 200.36 | 847 |
| | w/o | 29.05 | 566.75 | 2396 | 31.22 | 283.40 | 1198 |
| 16 | LQM | 28.95 | 368.00 | 1555 | 31.25 | 216.36 | 914 |
| | w/o | 28.90 | 557.94 | 2359 | 31.36 | 285.49 | 1207 |
| | UQ | 29.17 | 461.07 | 1949 | 31.29 | 219.15 | 927 |
| | w/o | 29.14 | 564.32 | 2386 | 31.37 | 285.27 | 1206 |
| 18 | LQM | 28.86 | 369.58 | 1562 | 31.04 | 219.98 | 930 |
| | w/o | 28.86 | 567.71 | 2400 | 31.11 | 290.02 | 1226 |
| | UQ | 29.23 | 462.36 | 1954 | 31.36 | 224.39 | 949 |
| | w/o | 29.10 | 563.87 | 2384 | 31.28 | 281.87 | 1191 |

## A.6. Influence of Uniqueness

Our proposed primitive position uniqueness method is for compatible with the SOTA learning-based methods based on SparseConv. Discard this module and the results are shown in Table 7. "drjoshson" and "room" are used as test examples. To highlight the influence of uniqueness, the other compression operation, such as feature dimension reduction, feature quantization, primitive pruning, and downstream point cloud encoder are disabled. We train 30000 epochs as vanilla 3DGS. We see that for most test conditions, disabling this module will incur significant primitive number and data size growth, while close PSNR with enable this module. For example, by disabling this module, LQM and UQ generate 1202K and 1143K primitives for "room" given 10 BD, which is almost 3 to 4 × data sizes compared to enable this module. Consequently, it reveals that for 3DGS, more primitives do not necessarily lead to better reconstruction quality. The primitive densification method proposed in vanilla 3DGS generation can be further improved if the data size is one of reference.

## A.7. Different Point Cloud Encoders on Position

To analyze the results of using 3DGS primitive position as the input of SOTA point cloud encoder, we select GPCC v23 (WG7, 2023) and SparsePCGC (Wang et al., 2022) as representations. Nine BDs: BD = 10 to 18, are tested to collect bitrate with lossless module. We focus on primitive position in this section, therefore, pruning and the compression operation related to other features are disabled. We train 30000 epochs as vanilla 3DGS. The curves of PSNR vs. bitrate of "bicycle" and "truck" are shown in Figure 11. "3DGS+UQ" means training a vanilla 3DGS then using UQ to quantize position as postprocessing, "LQM" means using the method proposed in Section 3.1.2 to generate primitive position, and "UQ" means using the feature quantization strategy proposed in Section 3.1.1 via UQ for primitive position.

We see that: 1) for the same bitrate, LQM and UQ report higher PSNR than 3DGS+UQ, indicating that introducing quantization into the generation of 3DGS is an effective strategy for reducing information loss; 2) for 3DGS+UQ, GPCC shows higher reconstruction quality than SparsePCGC. The reason is that SparsePCGC is based on SparseConv, which cannot deal with the case where multiple points are located at the same coordinate. A default duplicated point removal operator is performed before data compression, leading to loss of 3DGS quality. GPCC has two different settings considering duplicated points, i.e., keeping or removing, therefore, it can realize "real lossless" for 3DGS; 3) LQM and UQ can generate 3DGS without duplicated primitive position, facilitating SparsePCGC realizes "real lossless" 3DGS compression; 4) for the same bitrate, LQM reports slightly better performance than UQ. Besides, LQM can generate positions that do not need de-quantization before rendering, which means LQM can further save a position de-quantization time for real-time applications.

## A.8. Overall and Per Frame Results

Here we report overall and per frame results of HybridGS on Deep blending, tanks&temples, and Mip-NeRF360. We can see that: 1) for overall quality, HybridGS has 0.5-1.5 dB PSNR loss compared with the SOTA generative compression methods; 2) and a better quality can be reached by increasing the dimension of latent features. Some demo videos are available at `https:`

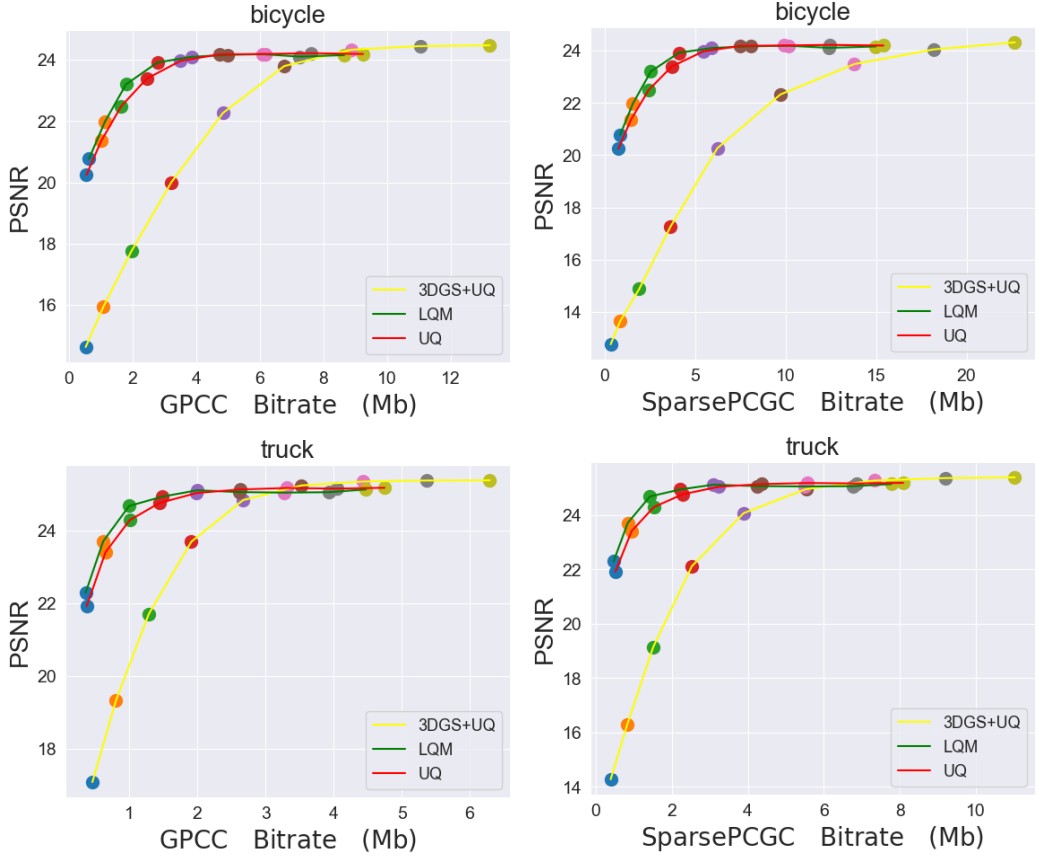

*Figure 11.* PSNR vs. bitrate curves of GPCC and PCGC.

//drive.google.com/drive/folders/14KIzFDIPSPdrKpXjtFh1HYUtG-E0Zs5W?usp=sharing.

*Table 8.* Overall results

| Dataset | | Tank&Temple | | | Deep Blending | | | MipNeRF360 | | |
|---|---|---|---|---|---|---|---|---|---|---|
| Method | | PSNR | SIZE | FPS | PSNR | SIZE | FPS | PSNR | SIZE | FPS |
| 3DGS-30K | | 23.14 | 411.00 | 154 | 29.41 | 676.00 | 137 | 27.21 | 734.00 | 134 |
| HybridGS | HR | 22.90 | 8.85 | 207 | 28.51 | 11.52 | 201 | 25.64 | 15.82 | 199 |
| kc=3, kr=2 | LR | 22.66 | 4.27 | 247 | 28.32 | 5.59 | 223 | 25.40 | 7.63 | 220 |
| HybridGS | HR | 23.12 | 11.10 | 195 | 29.05 | 16.35 | 191 | 25.97 | 21.73 | 189 |
| kc=6, kr=2 | LR | 22.83 | 5.27 | 214 | 28.82 | 7.92 | 212 | 25.75 | 10.47 | 210 |

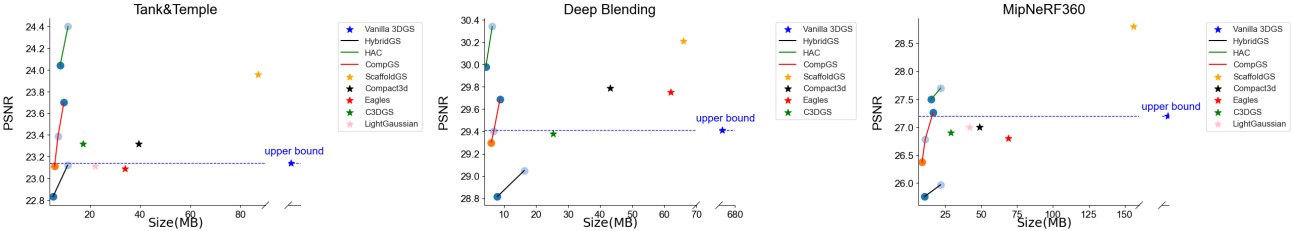

*Figure 12.* Overall RD curve.

*Table 9.* Tanks&Temples per scene results.

| Scene | Method | | Training PSNR | Testing PSNR | Size (MB) |
|-------|--------|---|---------------|--------------|-----------|
| truck | GS | | 27.50 | 25.39 | 611.58 |
| | HybridGS | HR | 26.96 | 24.53 | 13.50 |
| | $k_c = 3, k_r = 2$ | LR | 26.66 | 24.35 | 6.50 |
| | HybridGS | HR | 27.19 | 24.75 | 16.56 |
| | $k_c = 6, k_r = 2$ | LR | 26.90 | 24.62 | 7.92 |
| train | GS | | 25.37 | 21.89 | 256.73 |
| | HybridGS | HR | 24.42 | 21.26 | 4.19 |
| | $k_c = 3, k_r = 2$ | LR | 23.58 | 20.96 | 2.03 |
| | HybridGS | HR | 24.86 | 21.49 | 5.63 |
| | $k_c = 6, k_r = 2$ | LR | 24.01 | 21.04 | 2.62 |

*Figure 13.* Snapshots of HybridGS ($k_c = 3, k_r = 2$) on Tanks&Temples.

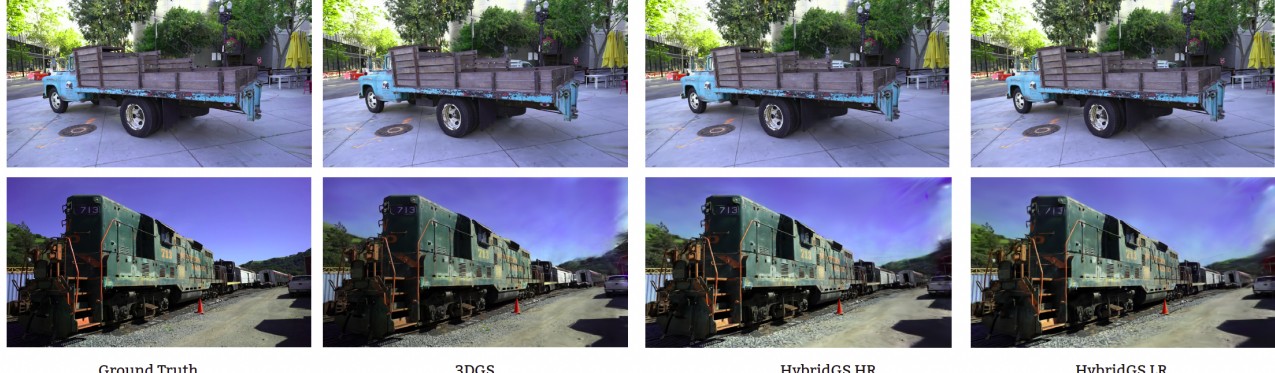

Ground Truth     3DGS     HybridGS HR     HybridGS LR

*Table 10.* Deep Blending per scene results.

| Scene | Method | | Training PSNR | Testing PSNR | Size (MB) |
|-------|--------|---|---------------|--------------|-----------|
| playroom | GS | | 37.63 | 30.03 | 550.67 |
| | HybridGS | HR | 33.52 | 29.89 | 12.15 |
| | $k_c = 3, k_r = 2$ | LR | 32.40 | 29.49 | 5.88 |
| | HybridGS | HR | 33.81 | 29.89 | 16.08 |
| | $k_c = 6, k_r = 2$ | LR | 33.11 | 29.68 | 7.79 |
| drjohnson | GS | | 36.21 | 29.06 | 779.93 |
| | HybridGS | HR | 32.01 | 27.12 | 10.88 |
| | $k_c = 3, k_r = 2$ | LR | 31.22 | 27.15 | 5.29 |
| | HybridGS | HR | 33.50 | 28.21 | 16.62 |
| | $k_c = 6, k_r = 2$ | LR | 31.86 | 27.95 | 8.04 |

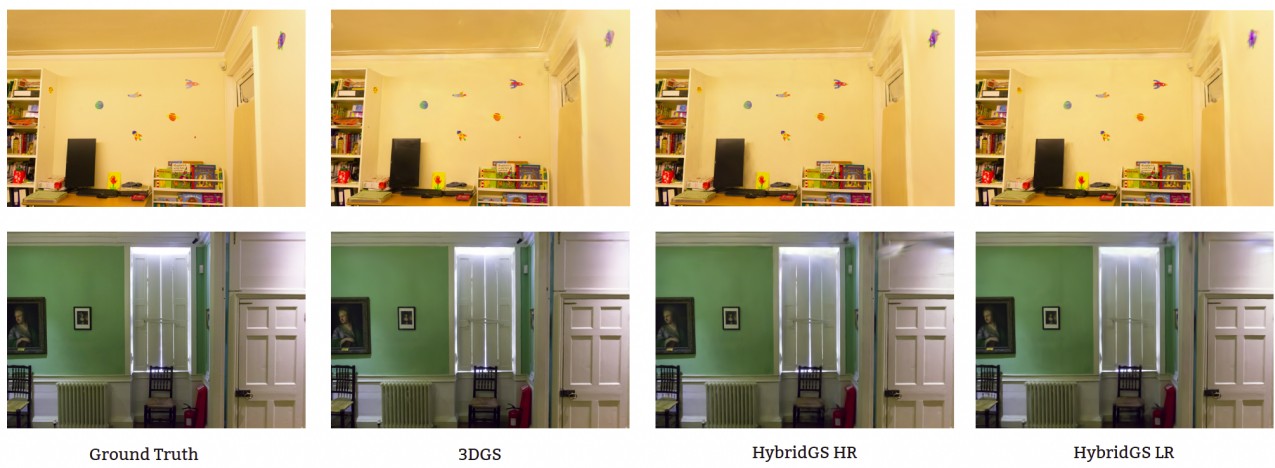

*Figure 14.* Snapshots of HybridGS ($k_c = 3, k_r = 2$) on Deep Blending.

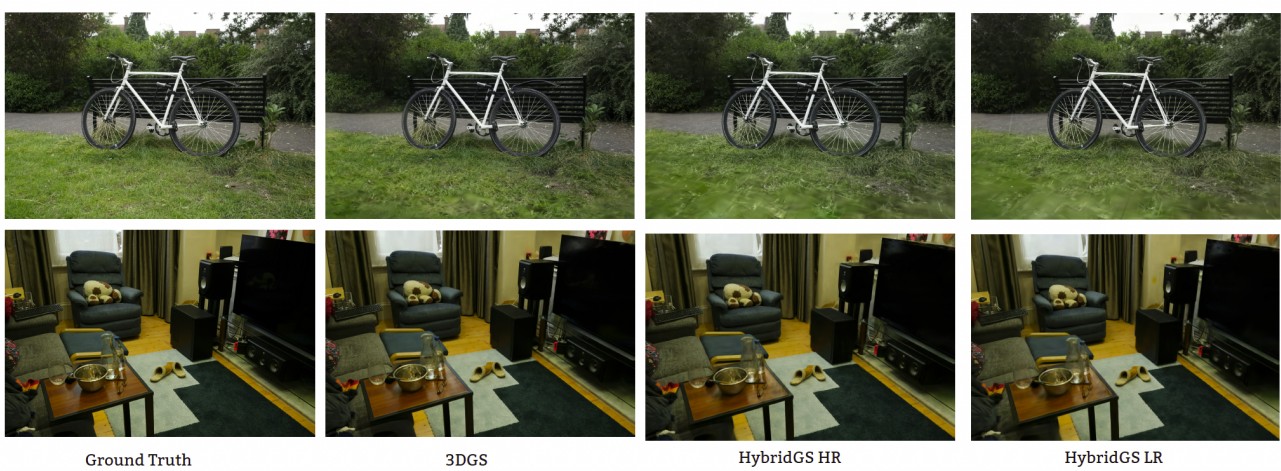

*Figure 15.* Snapshots of HybridGS ($k_c = 3, k_r = 2$) on MipNeRF360.

*Table 11.* MipNeRF360 per frame results.

| Scene | Method | | Training PSNR | Testing PSNR | Size (MB) |
|---|---|---|---|---|---|
| bicycle | GS | | 24.82 | 24.45 | 1443.84 |
| | HybridGS | HR | 21.03 | 24.08 | 22.88 |
| | $k_c = 3, k_r = 2$ | LR | 20.51 | 23.53 | 11.01 |
| | HybridGS | HR | 21.36 | 24.10 | 30.21 |
| | $k_c = 6, k_r = 2$ | LR | 21.34 | 23.76 | 14.52 |
| bonsai | GS | | 33.44 | 32.28 | 296.75 |
| | HybridGS | HR | 29.35 | 29.61 | 7.48 |
| | $k_c = 3, k_r = 2$ | LR | 28.44 | 29.25 | 3.62 |
| | HybridGS | HR | 30.65 | 30.86 | 11.62 |
| | $k_c = 6, k_r = 2$ | LR | 29.66 | 30.35 | 5.62 |
| counter | GS | | 30.46 | 29.07 | 284.30 |
| | HybridGS | HR | 28.98 | 26.88 | 7.07 |
| | $k_c = 3, k_r = 2$ | LR | 28.11 | 26.76 | 3.43 |
| | HybridGS | HR | 29.50 | 27.16 | 7.67 |
| | $k_c = 6, k_r = 2$ | LR | 28.26 | 27.02 | 3.72 |
| flowers | GS | | 23.22 | 21.27 | 856.45 |
| | HybridGS | HR | 21.32 | 20.25 | 17.21 |
| | $k_c = 3, k_r = 2$ | LR | 20.90 | 20.13 | 8.27 |
| | HybridGS | HR | 21.89 | 20.50 | 22.51 |
| | $k_c = 6, k_r = 2$ | LR | 21.41 | 20.33 | 10.78 |
| garden | GS | | 28.29 | 26.33 | 1392.64 |
| | HybridGS | HR | 26.80 | 26.37 | 36.13 |
| | $k_c = 3, k_r = 2$ | LR | 25.31 | 26.09 | 17.40 |
| | HybridGS | HR | 26.85 | 26.44 | 42.49 |
| | $k_c = 6, k_r = 2$ | LR | 25.23 | 26.10 | 20.42 |
| kitchen | GS | | 33.13 | 31.38 | 26.33 |
| | HybridGS | HR | 27.20 | 27.07 | 9.76 |
| | $k_c = 3, k_r = 2$ | LR | 26.75 | 27.04 | 4.70 |
| | HybridGS | HR | 27.78 | 27.56 | 12.29 |
| | $k_c = 6, k_r = 2$ | LR | 27.16 | 27.55 | 5.95 |
| room | GS | | 34.40 | 31.55 | 370.14 |
| | HybridGS | HR | 31.95 | 29.52 | 6.43 |
| | $k_c = 3, k_r = 2$ | LR | 31.04 | 29.23 | 3.14 |
| | HybridGS | HR | 32.38 | 29.75 | 8.28 |
| | $k_c = 6, k_r = 2$ | LR | 31.40 | 29.61 | 4.00 |
| stump | GS | | 29.76 | 26.24 | 1157.12 |
| | HybridGS | HR | 25.59 | 24.92 | 18.24 |
| | $k_c = 3, k_r = 2$ | LR | 24.76 | 24.67 | 8.83 |
| | HybridGS | HR | 26.19 | 25.09 | 35.00 |
| | $k_c = 6, k_r = 2$ | LR | 25.82 | 25.02 | 16.78 |
| treehill | GS | | 23.44 | 22.23 | 890.63 |
| | HybridGS | HR | 20.80 | 22.05 | 17.22 |
| | $k_c = 3, k_r = 2$ | LR | 20.63 | 21.91 | 8.29 |
| | HybridGS | HR | 21.67 | 22.24 | 25.51 |
| | $k_c = 6, k_r = 2$ | LR | 20.78 | 22.04 | 12.41 |

