# OpenReview forum: "HybridGS: High-Efficiency Gaussian Splatting Data Compression using Dual-Channel Sparse Representation and Point Cloud Encoder"
_ICML.cc/2025/Conference — ICML 2025 poster_

### Official Review · Reviewer_jCTH · 2025-03-13

**Overall Recommendation:** 3

**Summary:**

This paper proposes a new 3D Gaussian Splatting (3DGS) compression framework, HybridGS, which combines the advantages of generative and traditional compression methods. It improves the encoding and decoding speeds while ensuring the reconstruction performance.

**Claims And Evidence:**

Please see Other Strengths And Weaknesses.

**Essential References Not Discussed:**

N/A

**Experimental Designs Or Analyses:**

Please see Other Strengths And Weaknesses.

**Methods And Evaluation Criteria:**

Please see Other Strengths And Weaknesses.

**Other Comments Or Suggestions:**

I suggest that the author supplement BD-Rate and BD-PSNR for better analysis of experimental results.

**Other Strengths And Weaknesses:**

Strength

(1)The idea of this article is very novel. The author cleverly links 3DGS compression with point cloud compression, and realizes data format control and conversion through PAT-Q and DAT-R. This method can effectively take advantage of advanced point cloud codecs and greatly improve the compression efficiency of 3DGS. In addition, the author has simultaneously attempted to use G-PCC (traditional) and SparsePCGC (deep learning-based) as point cloud codecs for compression, which shows better generalization ability.

(2)The codec time of this framework is significantly better than that of previous methods, and it is easier to deploy the model in practice.

(3)The subjective effects presented in the supplementary materials demonstrate the excellent performance of this method.


Weakness

(1)The author's analysis of the experimental performance is insufficient. For example, from Table 1, it is difficult to analyze the actual compression effects of each method. It is recommended that the author use the commonly used BD-Rate and BD-PSNR metrics in the compression field to intuitively reflect the performance gaps among different methods.

(2)As the author mentioned in the "Limitations" section, the research on rate control in this article is not comprehensive enough. Generally, it is necessary to calculate the dependency relationship between the bit-rate and various quality control hyperparameters. This could be one of the future improvement directions for this work.

(3)The author did not include the reference code for this project in the supplementary materials, which has had a certain impact on the reproducibility of this thesis.

**Questions For Authors:**

N/A

**Relation To Broader Scientific Literature:**

N/A

**Theoretical Claims:**

Please see Other Strengths And Weaknesses.

---

> ### Author Rebuttal · Authors · 2025-03-30
>
> 1). Insufficient experiments
>
> We would like to thank you for positive recommendations. We provide some new results on BD-Rate and PSNR to compare the performance of the methods in consideration. They can be found via
> https://drive.google.com/drive/folders/1V1mxZq1IPXz2H0kF6_a7IsP8_iGOLCUu?usp=sharing.
> Generally, our method has around 0.5-1.5dB PSNR loss compared with HAC and CompGS with the same bitrate. However, the motivation of this paper is to realize a balance between compression ratio and coding time. We reduce coding time from tens of seconds or more than 1 minute to around 0.4s to 1.6s, which is the main advantage of this method.
>
> 2). Limitations on the current rate control scheme
>
> Thanks for your insightful comments. We agree with the reviewer that there are more parameters that can influence the rate control. For example, one factor of interest is the latent feature dimension during training. Different from reducing the bit depth and removing some primitive, decreasing the latent feature dimension during GS generation can lead to feature space collapse. It also requires training a new decoder, which will induce extra training time. How to achieve smooth latent feature dimension reduction without significantly affecting the rendering quality is thus a research topic worthy of investigation. We shall look into this problem in the future.
>
> 3). Code availability
>
> We have uploaded the code to GitHub. Due to the double-blind reviewer policy, we currently set it as a privacy repository. We will be more than happy to share it with the whole community after the paper is accepted.

---

> > ### Comment · Reviewer_jCTH · 2025-04-04
> >
> > Thanks for your reply. I decide to keep my score.

---

> > > ### Author Response · Authors · 2025-04-04
> > >
> > > We sincerely appreciate your support and the time you dedicated to reviewing our work!

---

### Official Review · Reviewer_15YA · 2025-03-13

**Overall Recommendation:** 2

**Summary:**

In this work, the authors propose a compression framework for 3DGS. A lightweight decoder $D$ composed of a one hidden layer MLP is introduced to compress original high-dimensional GS features into Low-dimensional latent features $f$  for quantization and compression, where the rate is controlled by adjusting the number of GS primitives or quantization depths.

**Claims And Evidence:**

NA.

**Essential References Not Discussed:**

No.

**Experimental Designs Or Analyses:**

Some experiments may be missing, please check the strengths/weaknesses section.

**Methods And Evaluation Criteria:**

The benchmarks may be reasonable.

**Other Comments Or Suggestions:**

Please check the typos in this work,e.g., Diffenert in Fig.1.

**Other Strengths And Weaknesses:**

Strengths

(1) The authors proposed a 3DGS compression framework without 3DGS optimization;

(2) The proposed method has higher compression efficiency than baselines.

Weaknesses

(1) For me, the whole framework is hard to understand. I would suggest the authors to re-write the papers and add clearer diagrams to demonstrate this method. The confusing presentation of this work may make it hard to be accepted as a ICML submission.

(2) The performances of the proposed method shows quite inferior performances compared to existing methods like ComGS, and HAC. Although some improvements show up in the coding time as shown in Table. 2, the dropping of performances cannot be ignored;

(3) I don't quite get the dividing of generated samples into subsamples. Does it mean that we compress multiple point clouds attached with different GS attributes repeatedly?

(4) There is not any qualitative comparisons in this paper, which makes it difficult to intuitively judge the performance differences.

**Questions For Authors:**

Please check the strengths/weaknesses section.

**Relation To Broader Scientific Literature:**

No.

**Theoretical Claims:**

This work does not include theoretical claims needed to be proved.

---

> ### Author Rebuttal · Authors · 2025-03-30
>
> 1). Paper writing
>
> We would like to apologize for not motivating this work well and presenting our method clearly. In the revised paper, we shall replace Fig2 with a better illustrated one given in
> https://drive.google.com/drive/folders/1CwMbhm4l44oXD5MnCP3slbJHgw_vZ48c?usp=sharing.
>
> Besides, we will include the following arguments with proper literature citations to better motivating this work:
>
> a). Current generative GS compression methods, such as HAC and CompGS, provide impressive compression ratios but at the cost of very slow encoding and decoding speed, due to their use of implicit 3DGS bitstreams.
>
> b). For the scenarios of visual media streaming, it has been shown that users start to feel dissatisfaction when the latency is larger than 400ms. Furthermore, 1s is about the limit for the user’s flow of thought to stay uninterrupted, and 10s is the limit for keeping the user’s attention focused on the dialogue. Therefore, the quality of experience (QoE) will drop sharply if it takes about 1 min to decode the GS from the bitstream.
>
> c). For dynamic 3DGS content, the decoding speed will also affect the FPS and content fluency.
>
> Third, we shall completely rewrite the last three paragraphs of the Introduction section. They will now cover the following aspects:
>
> a). To realize fast 3DGS coding and decoding, we propose to generate compact 3DGS in the explicit domain instead of the implicit domain, which enables the use of reliable and effective point cloud encoders to generate bitstream.
>
> b). For explicit and compact 3DGS generation, a dual-channel sparse representation is adopted to reduce the data volume. This reduces the computation burden of downstream point cloud encoders and has less compression-related distortion. The first channel produces sparse 3DGS primitive distribution, where a learnable quantizer-based method (LQM) is employed to obtain dequantization-free primitive positions. The second channel generates sparse features for each primitive. Feature dimension reduction and a robust quantizer (RQ) are utilized to find low-rank primitive features.
>
> c). With the same bitrate, HybridGS suffers from around 0.5-1.5dB loss in PSNR, compared with SOTA generative compression methods. But using HAC as a baseline, we decrease the encoding and decoding latency from tens of seconds or more than 1 minute to around 0.4s to 1.6s. This clearly demonstrates the practicality of the proposed framework.
>
> 2). Compression ratio
>
> Thanks very much for the comment. Actually, in the “Conclusion Limitations” Section of the paper, we summarized that the compression ratio of the proposed HybridGS is lower than that of some end-to-end methods. As discussed in our response to your previous comment, this work aimed at achieving a better balance between compression ratio and the complexity. HybridGS is among the few existing work that successfully reduces the coding time to a level potentially meeting the requirement of use cases such as GS-based dynamic content delivery and streaming. For more explanation about why coding speed is important please refer to the first question of reviewer nV7f. It also has rooms for further performance improvements through integrating e.g., 3DGS quality optimization and new point cloud encoders.
>
> 3). Division of generated samples into subsamples
>
> Thanks for the question. Please allow us to clarify. The purpose of dividing generated samples into subsamples is to make them compatible with point cloud encoders. As mentioned in Section 3.2.1 of the paper, current point cloud encoders support data format with “xyzrgb” or “xyzf” only, because point clouds generally do not have as many feature channels as 3DGS. Considering that the point cloud shares the same data format with 3DGS, we can have two different methods for using existing point cloud encoders to compress 3DGS data.
>
> The first approach is to divide 3DGS data into subsamples. For example, vanilla GS has x, y, z, r, g, b, sh1, sh2, …, sh45, opacity, scaling1, scaling 2, scaling3, rotation1, …, rotation4. They can be grouped as sample1: x, y, z, r, g, b; sample2: x, y, z, sh1, sh2, sh3, etc. These subsamples can be directly input to point cloud encoders. This does mean we need to call the GPCC encoders multiple times though.
>
> The second method is to modify current point cloud encoders to make them be able to handle the one-time 3DGS sample loading. This approach can further exploit OpenMP to process GS attributes in parallel, leading to even faster coding time. We do not follow this way because compared with the first method, it will has same the compression ratio.
>
> 4). Qualitative comparisons
>
> Qualitative results are given in Appendix A9, where the snapshots of compressed 3DGS samples are shown. We also provide average performance on three datasets in the first question of reviewer rykQ. Additional videos for more illustrations are now available via
> https://drive.google.com/drive/folders/1YaFvkHLDQ10CAV0NLGmT9KhQitBnrdIU?usp=sharing.

---

### Official Review · Reviewer_nV7f · 2025-03-14

**Overall Recommendation:** 3

**Summary:**

This paper aims to compress 3d Gaussians into very small sizes for storage efficiency. The core idea of the proposed HybridGS is to combine traditional point cloud compression method and the generative coding compression method. The most advantage of HybridGS compared to previous 3DGS compression method is it requires less encoding and decoding time.

**Claims And Evidence:**

Yes. This paper claims that the proposed method can compress 3DGS at a very fast speed, which is verified by subsequent experimental results in table 2.

**Essential References Not Discussed:**

No.

**Experimental Designs Or Analyses:**

Yes. In table 2, the proposed method achieves significantly better compress efficiency compared to other methods.

**Methods And Evaluation Criteria:**

Yes. Five popular scenes from four datasets (deep blending, tanks&temples, Mip-NeRF360, PKU-DyMVHumans) are included in the comparisons. And some more scenes from deep blending and Mip-NeRF360 are provided in the appendix.

**Other Comments Or Suggestions:**

1. It is hard for me to understand the components column in table 3. And there seems to be something wrong with the spacing of the table 2 and 3.

**Other Strengths And Weaknesses:**

Strengths:
1. The proposed method can compress 3DGS at a fast speed, which is different from previous methods.
2. This paper provides extensive experiments on various datasets.

Weaknesses:
1. The rendering quality is worse than other 3DGS compression methods such as HAC and CompGS as shown in table 1.

**Questions For Authors:**

1. I am not very sure about if reducing the encoding and decoding time of 3DGS compression really necessary. I would consider raising the score if it is clearer why compression speed is important.

**Relation To Broader Scientific Literature:**

The feature channel compression in Dual-Channel Sparse Representation is widely used a lot of previous studies such as Scaffold-GS, LightGaussian.

**Theoretical Claims:**

Yes. the Dual-Channel Sparse Representation and High-Efficiency Coding. The two components make scenes.

---

> ### Author Rebuttal · Authors · 2025-03-30
>
> 1). Importance of compression speed
>
> Thanks very much for your comments on the importance of compression speed. Please allow us to clarify.
>
> The processing latency of visual media, especially the encoding and decoding time  [R1], has become an essential utility factor, considering that 5G network can provide larger bandwidth and network transmission is now much faster than that in past decades [R2]. As an emerging type of visual media, GS has many streaming use cases [R3], where the encoding and decoding speed sometimes is even more important than the compression ratio itself. According to International Telecommunication Union (ITU) [R4], a latency higher than 400ms will result in user dissatisfaction. For highly interactive scenarios, this threshold drops to 250ms. The findings from studying the response time [R5], and empirical video-on-demand and video live streaming cases indicate that 1s is probably the limit for the user’s flow of thought to stay uninterrupted, and 10s is the limit for keeping the user’s attention focused on the dialogue. As a result, it can be expected that the quality of experience (QoE) will be greatly degraded if it takes close to 1 minute to decode the GS from the received bitstream.
>
> Current 3DGS compression work, on the other hand, mostly concentrate on achieving high compression ratio, without paying sufficient attention to the coding speed. Therefore, there exists a clear gap between the academic and industrial needs for lightweight encoding and decoding schemes, and the current literature and practice. We hope our work will bridge this gap to some extent. This is also the reason behind our decision of not including any optimization techniques for improving the delivery quality.
>
> We shall include these discussions in the revised paper to address your comments.
>
> [R1] Kim et al., "C3: High-performance and low-complexity neural compression from a single image or video," Proceedings of the IEEE/CVF Conference on Computer Vision and Pattern Recognition (CVPR), 2024.
>
> [R2] Wang et al., "Inferring end-to-end latency in live videos," IEEE Transactions on Broadcasting 68.2 (2021): 517-529.
>
> [R3] Sun et al., "3DGStream: On-the-fly training of 3D Gaussians for efficient streaming of photo-realistic free-viewpoint videos," Proceedings of the IEEE/CVF Conference on Computer Vision and Pattern Recognition (CVPR), 2024.
>
> [R4] Recommendation ITU-T G.1051, "Series G: Transmission systems and media, digital systems and networks - Multimedia quality of service and performance - Generic and use-related aspects," 2023
>
> [R5] J. Nielsen. Usability Engineering. Morgan Kaufmann, 1994.
>
> 2). Rendering quality
>
> Thanks for the insightful comments. The rendering quality of the proposed HybridGS has an upper bound, which is the quality of the vanilla 3DGS. This is shown in Table 1 and Figure 5. As an example, the vanilla GS `bicycle’ has a PSNR of 24.49dB, while HybridGS provides a PSNR of 24.10dB. In our experiments, we set the latent dimension of color to be 3 or 6. Increasing the dimensionality of color and rotation latent features and/or applying a larger bit depth would lead to improved PSNR results, better approaching the performance of vanilla 3DGS. For example, if we increase the dimension of color latent features to 9 while keeping other settings unchanged, HybridGS can now provide a PSNR of 24.25dB for ‘bicycle’. On the other hand, HAC reports a PSNR of 25.00dB for ‘bicycle’, a quality level that even requires introducing techniques to improve the quality of the vanilla 3DGS. As pointed out in the answer to your previous comment on the importance of compression speed, we did not include any optimization techniques in this work. But the design of HybridGS does allow additional quality enhancement method to be incorporated, which is an important direction for future research.
>
> 3). Table 3
>
> We would like to apologize for this clarity issue. Table 3 in the paper gives the size of different GS features before and after compression via GPCC. It illustrates the bitrate distribution of the proposed HybridGS. In fact, HybridGS first generates a compact and explicit GS file (.ply file) that consists of primitive position (xyz), color latent feature, opacity, scaling, and rotation latent features. These data can be considered as point cloud data and thus can be further compressed by GPCC, with each of them having an individual bitstream. Take ‘bicycle’ as an example. In Table 3, ‘position 2.72 MB (7.36)’ means that before applying GPCC, the size of the GS primitive position information is 7.36 MB, while after using GPCC, the bitstream size reduces to 2.72 MB. This indicates around 2.7x lossless compression. After the GPCC compression, the bitstreams of the aforementioned data, as well as metadata and the parameters of two small MLPs, will be stored or used for streaming. They form the final output bitstream.
>
> In the revised paper, we shall add more space between Tables 2 and 3 for better clarity.

---

> > ### Comment · Reviewer_nV7f · 2025-04-02
> >
> > While it is sill confusing to me. I understand the importance of rendering speed 3dgs. But why do we need to compress them fast? The optimization of 3DGS from multi-view images takes a significant amount of time. Compared to this, is the time overhead of different compression methods negligible? From this perspective, I don’t understand why the compression time overhead is considered important.

---

> > > ### Author Response · Authors · 2025-04-03
> > >
> > > Thanks for acknowledging reading our response to your outstanding comments.
> > >
> > > We agree that the coding overhead consists of two parts, namely the encoding time and decoding time. The reviewer is correct that the encoding time can be neglected due to current 3DGS optimization already taking significant amount of time. The decoding speed is much more important and meaningful, as it influences greatly the quality of experience (QoE) at the user side: after the user requests certain content and receives bitstream from the provider, they need to decode it from bitstream first, and then view 3DGS.
> > >
> > > One main contribution of our work is thus on the evident reduction of the 3DGS decoding time provided by the proposed HybridGS coding scheme. Extensive empirical study has confirmed this. As an example, in Table 2, for ‘bicycle’, the decoding time of HAC is 80.09s, while our method only needs 1.77s.
> > >
> > > Besides the clear improvement in the decoding speed, HybridGS offers a comparable compression ratio, which could help decrease the memory cost for 3DGS data storge as well as the bandwidth requirement for 3DGS data transfer.
> > >
> > > Another evidence in support of the importance of our work comes from industry. In the ongoing MPEG 150th meeting in April 2025, Qualcomm, Samsung, Bytedance, and Xiaomi submitted a joint proposal “m72430 [GSC][JEE6.4-related] On the use case and requirements for lightweight GSC” [1]. Here, GSC stands for Gaussian Splatting Compression, which is actually an Adhoc group of MPEG WG4 to explore GS compression standardization. Here, the requirements for “Low complexity/low-power encoding and decoding” were highlighted and “Fast frame encoding and decoding” were also mentioned.
> > >
> > > [1] ISO/IEC JTC 1/SC 29/WG 4 m72430, “[GSC][JEE6.4-related] On the use case and requirements for lightweight GSC”. 2025

---

### Official Review · Reviewer_BMP7 · 2025-03-18

**Overall Recommendation:** 3

**Summary:**

HybridGS aims at the data compression of 3DGS. It takes advantage of both generative compression technique and traditional compression technique by first generating a compact explicit 3DGS representation and then encoding it with a standard point cloud codec​. It achieves a higher encoding and decoding speed compared to other generative compression methods. Its key innovations include: (1) Dual-Channel Sparse Representation; (2) Rate Control Scheme: progressively prune more primitives to reduce point count, and/or lower the quantization precision of features (bit-depth) uniformly across attributes​.

## update after rebuttal
After reviewing the rebuttal, Most of my concerns have been addressed, so I maintain my score.

**Claims And Evidence:**

The major claims in the submission is the higher encoding and decoding speed for Gaussian data compression with comparable rendering performance. This claim is supported by clear comparison in the experiments.

**Essential References Not Discussed:**

The references are adequate. However, as one of the highly related work in this paper, the reference info of lightGaussian is out-of-date. It would be better to update it. (It has been accpected by NIPS2024, but the paper still cites its arxiv version.)

**Experimental Designs Or Analyses:**

The experiments are sound.

I appreciate that the authors list the specific results of each scenario in the supplementary materials, but it would be better to have a complete comparison in Mipnerf360 in the main paper, which will make it easier for other researchers in the community to compare different works.

**Methods And Evaluation Criteria:**

The evaluation criteria is adequate. However, the FPS(rendering speed) is better to be involved.

**Other Comments Or Suggestions:**

As a rendering-related work, I strongly suggest that the author should include video results after compressing.

My current score is more like a boardline. For me, the biggest problem with this paper is the writing. Although I am familiar with 3DGS and some compression work, I still need to spend some time to get the core contribution of this paper. For example, in the Introduction section, the author spent a lot of time introducing the specific implementation process of the method, which makes it difficult for readers to understand how the focus of this article is to achieve the acceleration of encoding and decoding time.

**Other Strengths And Weaknesses:**

Strengths:
1. HybridGS offers a practical solution for Gaussian Compression.
2. The processing(encoding and decoding) speed is much higher.
3. The evaluation is adequate.

Weaknesses:
1. While HybridGS is close to SOTA, it doesn’t surpass the best neural methods in pure compression efficiency. HAC or CompGS can often reach smaller bitrates at the cost of encoding time.
2. The method with combined compression tecnique is somewhat complex to implement compared to a purely learned approach.
3. HybridGS outputs an explicit point cloud plus small MLPs. The memory required to hold the decoded point cloud in memory could be larger than for a fully implicit decoder.

**Questions For Authors:**

See above

**Relation To Broader Scientific Literature:**

This paper combines the technique in neural rendering compression and point cloud compression, which brings the compression technique to 3D Gaussian Splatting(3DGS). It could benefit the 3DGS community, but the broader impact is limited.

**Theoretical Claims:**

The paper is largely an empirical study, so no formal proofs to verify. However, it does introduce some theoretical reasoning. They present two optimization formulations (8) and (9) for rate control under bandwidth constraint​, which are consistent with known practices.

---

> ### Author Rebuttal · Authors · 2025-03-28
>
> 1). FPS (rendering speed), averaged and complete results
>
> Thanks very much for your suggestions. Please refer to our response to Question 1 of Reviewer rykQ for complete results and newly generated FPS results over different datasets.
>
> 2). Essential references
>
> We shall update the reference information of the lightGaussian work in the revised paper. We would like to apologize for not using the latest information.
>
> 3). Compression ratio
>
> In the “Conclusion Limitations” Section of the paper, we admitted that the compression ratio of the proposed HybridGS method is in fact lower than some end-to-end methods. However, the goal of this work is to achieve a better balance between compression ratio and coding speed. More on the motivation for putting efforts to reduce the coding time can be found in our response to Question 1 of reviewer nV7F. For some use cases, coding speed is of great importance, such as dynamic content delivery and streaming, where existing methods can hardly meet the high-speed 3DGS encoding and decoding requirements. The results in Table 2 of the paper showed that with HybridGS, we can reduce the encoding and decoding time from tens of seconds or over 1 minute to 0.4s to 1.6s. This lays down a nice starting point for future researches on real-time GS coding and decoding.
>
> 4). Implementation complexity
>
> Thanks for the comments. HybridGS has two steps. The first step is to generate a compact GS file, which is then compressed by an existing point cloud encoder in the second step. To facilitate the understanding and applications of the proposed technique, we shall provide an easy-to-use script for realization. One advantage of our method is that it is compatible with many downstream point cloud encoders. In other words, you can integrate the latest encoder in a straightforward manner.
>
> 5). Memory consumption of HybridGS
>
> We agree with the reviewer that one disadvantage of keeping data explicit is that it requires a larger memory. Explicit data is more friendly for realizing fast coding speed, while implicit data is more friendly for reducing data size. In our opinion, compared with vanilla 3DGS, HybridGS itself incorporated a few techniques that can help to decrease the memory cost. For example, we introduced primitive uniqueness and pruning in HybridGS to reduce the number of generated GS primitives. In the future work, we shall consider how to further reduce the memory cost while keeping the characteristic of fast encoding and decoding speed.
>
> 6). Video rendering
>
> For illustration purpose, we have provided some snapshots in Appendix A.9. We shall show some video results. Due to time limitations, please check the link for some demo videos, more video results will be provided later. Due to the video coding format, VLC media player is recommended to play the video, other players might show some blur.
> https://drive.google.com/drive/folders/1YaFvkHLDQ10CAV0NLGmT9KhQitBnrdIU?usp=sharing
>
> 7). Paper writing
>
> We would like to apologize for not motivating this work well. In the revised paper, we shall rewrite the last several paragraphs of the Introduction section to reduce the technological details, while highlighting more the following motivations:
>
> (a) Encoding and decoding over explicit data is much faster than over implicit data. Therefore, in this paper, we choose to generate compact and explicit GS data file first, then followed by applying an existing point cloud encoder to generate the bitstream.
>
> (b) The volume of explicit data will influence coding time significantly. Therefore, we use LQM to produce sparse primitive distribution, and use feature dimension reduction to generate low-rank feature matrices. The size of explicit data to be compressed is thus reduced significantly.
>
> (c) We introduce quantization and uniqueness during the GS generation, which means practical encoding does not need any preprocessing like quantization. Besides, this also facilitates the reduction of encoding and decoding time.
>
> We re-plot Fig. 2 for better illustrating our method, which can be found in
> https://drive.google.com/drive/folders/1CwMbhm4l44oXD5MnCP3slbJHgw_vZ48c?usp=sharing

---

### Official Review · Reviewer_rykQ · 2025-03-24

**Overall Recommendation:** 3

**Summary:**

The authors propose HybridGS to compress 3D Gaussian splatting. The method first generates compact 3D Gaussians using dimension reduction, quantization of features, and positions. Then, it uses existing point cloud encoders to further compress the generated 3D Gaussians. The method achieves compression performance similar to SOTA methods with faster coding and decoding speeds.

## update after rebuttal
After reviewing the rebuttal, I feel that some of my concerns have been addressed, so I have decided to move the recommendation to weak accept.

**Claims And Evidence:**

The claims are supported.

**Essential References Not Discussed:**

No.

**Experimental Designs Or Analyses:**

In the main paper, the authors demonstrate their methods on five selected scenes. However, there is a lack of averaged metrics for the entire datasets. Although PSNR and SIZE of additional results are reported in the appendix, it would be more conclusive to include the average trend in Table 1 and Figure 5.

**Methods And Evaluation Criteria:**

Yes.

**Other Comments Or Suggestions:**

1. At the top of every page, the title line has become "Title Suppressed Due to Excessive Size."
2. Line 134, right column, "where ̄, Cov, and Var represent ...", the average operator symbol is above the comma, which looks confusing at first glance.

**Other Strengths And Weaknesses:**

Weakness:
1. The HybridGS method does not present enough novelty. The idea of using the existing point cloud encoders, such as G-PCC, has already been explored in GGSC [1]. The other components for dimension reduction and quantization do not provide enough novelty to be accepted at this venue.
2. The compression rate of the method is still lower than other end-to-end method [2].
3. Although the encoding and decoding time significantly decrease thanks to existing point cloud encoders, the computation time for dimension reduction, quantization, and pruning of the Gaussians is not discussed in detail.


[1] Yang, Q., Yang, K., Xing, Y., Xu, Y., and Li, Z. A benchmark for gaussian splatting compression and quality assessment study. In Proc. ACM Int. Conf. Multimedia in Asia. Association for Computing Machinery, 2024. doi: 10.1145/3696409.3700172.

[2] Liu, X., Wu, X., Zhang, P., Wang, S., Li, Z., and Kwong, S. Compgs: Efficient 3d scene representation via compressed gaussian splatting. arXiv preprint arXiv:2404.09458, 2024.

**Questions For Authors:**

No.

**Relation To Broader Scientific Literature:**

The method is a combination of learning-based quantization and dimension reduction methods, as well as signal-processing-based compression methods.

**Theoretical Claims:**

There is no proof to verify.

---

> ### Author Rebuttal · Authors · 2025-03-28
>
> 1).	Averaged metrics for the entire dataset
>
> We shall include a table and figures to show the averaged metrics for the entire dataset. The table is given below. The proposed HybridGS exhibits 0.5dB to 1.5dB loss in PSNR compared with HAC and CompGS under the same bitrate.
>
> The figures can be accessed via
> https://drive.google.com/drive/folders/1V1mxZq1IPXz2H0kF6_a7IsP8_iGOLCUu?usp=sharing .
>
> |                        |       | Tank&Temple | | | DeepBlending | | | MipNerf360 | | |
> | ---------------------- | ----- |:----------:|---|---|:------------:|---|---|:----------:|---|---|
> |                        |       | PSNR | SIZE  | FPS  |  PSNR | SIZE  | FPS  |  PSNR | SIZE  | FPS  |
> | 3DGS-30K               |       | 23.14 | 411.00 | 154 | 29.41 | 676.00 | 137 | 27.21 | 734.00 | 134 |
> | HybridGS kc=3, kr=2 | HR    | 22.90 | 8.85   | 207 | 28.51 | 11.52  | 201 | 25.64 | 15.82  | 199 |
> |                        | LR    | 22.66 | 4.27  | 247 | 28.32  | 5.59  | 223 | 25.40  | 7.63  | 219 |
> | HybridGS kc=6, kr=2 | HR    | 23.12 | 11.10  | 195 | 29.05 | 16.35  | 191 | 25.97 | 21.73  | 189 |
> |                        | LR    | 22.83 | 5.27  | 214 | 28.82  | 7.92  | 211 | 25.75  | 10.47 | 210 |
>
> 2). Novelty aspect
>
> We would like to apologize for not well motivating this work. We shall better address the motivation and importance of our work in the revised paper.
>
> We agree with the reviewer that both GGSC and HybridGS use existing point cloud encoders to realize data compression. But HybridGS adopted completely different strategies, making it superior to GGSC empirically.
>
> GGSC uses vanilla GS to generate GS samples, then carries out quantization to convert GS attributes into integers and finally employs GPCC to compress geometry. The disadvantages of this approach were discussed in the Introduction and Experiment sections of the original paper. They include
>
> (a) Using quantization as postprocessing introduces obvious distortion (see Fig. 11 in the Appendix). HybridGS integrates quantization into the GS generation rather than using it postprocessing, leading to a maximum of 4.5dB PSNR gain.
>
> (b) The quantization method of GGSC exhibits duplicated GS positions, which requires that the downstream point cloud encoder is able to handle duplicate points. This renders the use of SOTA point cloud encoders based on SparseConv infeasible as they take unique GS positions only (see Section 2.1). HybridGS achieves uniqueness during the GS generation to make it compatible with all existing point cloud encoders.
>
> (c) Vanilla GS produces a large volume of data. Applying point cloud encoders directly incurs very long encoding and decoding latency. As shown in Table 2, GGSC spends over 2 mins for “bicycle”. HybridGS tackles this problem using dimension reduction for achieving low-rank feature generation and an LQM module to induce sparse and dequantization-free geometry, decreasing the encoding and decoding time from tens of seconds or over a minute to 0.4s to 1.6s.
>
> The main contribution of this work, in our opinion, is to establish a framework capable of integrating the developed compression method as well as the rate control scheme with existing point cloud encoders to attain fast encoding/decoding while maintaining comparable compression ratio and quality. More justifications of the importance of reducing the encoding/decoding latency can be found in our response to the first question of Reviewer nV7F. They will be included in the revised paper.
>
> 3). Compression ratio
>
> As discussed above, the goal of this work is a better balance between the compression ratio and coding speed. In the “Conclusion Limitations” Section of the paper, we did admit that the compression ratio of HybridGS is lower than some end-to-end methods. We also did not include any optimization techniques for improving the quality either. The gain is significant improvement in the coding speed, which is important and meaningful for dynamic content delivery and streaming scenarios.
>
> 4). Computational complexity
>
> Thanks for the comment. The fast coding speed of HybridGS is due to using the point cloud codec as well as the sparse characteristic of the explicit GS data. On the other hand, the dimension reduction, quantization, and pruning during the GS generation requires marginal overhead. In a whole, the GS generation time of HybridGS is close to the vanilla GS under the same training epochs. As an example, for “dance” with 7k training epochs, HybridGS requires 6 mins while vanilla GS uses 5.6 mins. Another evidence on the low computation time for dimension reduction and quantization has been reported at the end of Section 4.2.2. After decoding, for the largest sample ``bicycle’’, the dimension reduction and quantization for color only takes 1s and 0.9s, while 0.6s and 0.001s for rotation.
>
> 5). Editing issues
>
> Thank you for your careful reading. We shall fix them all in the revision.

---

### Decision · Program_Chairs · 2025-05-01

**Decision:**

Accept (poster)

**Comment:**

The paper proposes a novel 3DGS compression framework HybridGS which integrates the strengths of both generative and traditional compression methods to reduce encoding and decoding times while maintaining comparable reconstruction quality to SOTA methods. The reviewers acknowledged several strengths of the paper, particularly its innovative approach of combining traditional point cloud compression techniques with generative coding methods, leading to faster encoding and decoding speeds. Reviewer nV7f emphasized this point, noting the practical implications of improved compression speeds.

During the rebuttal period, the authors provided solid explanation with additional experimental results and successfully addressed the concerns raised by the reviewers and therefore both Reviewer rykQ and Reviewer nV7f upgraded their ratings to weak accept.

Given that the paper presents a valuable concept and has the potential to advance the field of 3D compression, the AC recommends accepting this paper to ICML 2025. Congratulations! Please be aware that the authors are strongly encouraged to address reviewer’s concerns and enhance clarity (see Reviewer 15YA) in the camera-ready version.